# Visualizing atomic structure and magnetism of 2D magnetic insulators via tunneling through graphene

Zhizhan Qiu [1,8], Matthew Holwill [2,8], Thomas Olsen [3,8], Pin Lyu[1], Jing Li [4], Hanyan Fang [1], Huimin Yang[1], Mikhail Kashchenko[2,5], Kostya S. Novoselov[2,4,6,7 ✉] & Jiong Lu [1,4 ✉]

The discovery of two-dimensional (2D) magnetism combined with van der Waals (vdW) heterostructure engineering offers unprecedented opportunities for creating artificial magnetic structures with non-trivial magnetic textures. Further progress hinges on deep understanding of electronic and magnetic properties of 2D magnets at the atomic scale. Although local electronic properties can be probed by scanning tunneling microscopy/spectroscopy (STM/STS), its application to investigate 2D magnetic insulators remains elusive due to absence of a conducting path and their extreme air sensitivity. Here we demonstrate that few-layer $CrI_3$ (FL-$CrI_3$) covered by graphene can be characterized electronically and magnetically via STM by exploiting the transparency of graphene to tunneling electrons. STS reveals electronic structures of FL-$CrI_3$ including flat bands responsible for its magnetic state. AFM-to-FM transition of FL-$CrI_3$ can be visualized through the magnetic field dependent moiré contrast in the d$I$/d$V$ maps due to a change of the electronic hybridization between graphene and spin-polarised $CrI_3$ bands with different interlayer magnetic coupling. Our findings provide a general route to probe atomic-scale electronic and magnetic properties of 2D magnetic insulators for future spintronics and quantum technology applications.

---

[1] Department of Chemistry, National University of Singapore, 3 Science Drive 3, Singapore 117543, Singapore. [2] National Graphene Institute, University of Manchester, Manchester M13 9PL, UK. [3] Computational Atomic-scale Materials Design (CAMD), Department of Physics, Technical University of Denmark, 2800 Kgs, Lyngby, Denmark. [4] Centre for Advanced 2D Materials (CA2DM), National University of Singapore, 6 Science Drive 2, Singapore 117546, Singapore. [5] Center for Photonics and 2D Materials, Moscow Institute of Physics and Technology, Dolgoprudny 141700, Russia. [6] Department of Materials Science & Engineering, National University of Singapore, 9 Engineering Drive 1, Singapore 117575, Singapore. [7] Chongqing 2D Materials Institute, Liangjiang New Area, Chongqing 400714, China. [8] These authors contributed equally: Zhizhan Qiu, Matthew Holwill, Thomas Olsen. ✉email: kostya@nus.edu.sg; chmluj@nus.edu.sg

Scanning tunneling microscopy (STM) is a versatile tool when it comes to the study of electronic properties of metals at the atomic scale. Despite its obvious advantages, this technique also has a number of drawbacks: it can only investigate conductive materials and lacks direct access to the information about the momentum distribution and magnetic ordering of the electronic states. Recently, engineering vdW heterostructures promoted itself as a versatile tool to modify and study the electronic and magnetic structures of various 2D materials: insulators, semiconductors, metals, superconductors, and ferromagnets[1–5]. Here, we demonstrate that the application of vdW technology to the STM will dramatically expand the capabilities of the latter, allowing it to study insulating materials and gaining information about the magnetic ordering in 2D ferromagnets[6–9].

To this end, we assemble vdW heterostructures based on investigated 2D materials covered with monolayer graphene. Graphene, being conductive, ideally suits STM. At the same time, its low density of states (DOS) and the ability of its electronic states to hybridize with the electronic states from other 2D materials allow for gaining information about materials buried underneath at the atomic scale[10–13]. Furthermore, the projection of the electronic states of other materials on graphene depends strongly on the atomic arrangements; thus, additional information (like stacking between buried layers, or even information about magnetic structure) can be extracted from the close examination of the moiré structure between graphene and materials under study.

## Results

### Structural characterization of graphene/CrI3/graphite.

Here, we used STM to study mechanically exfoliated FL-CrI3 sandwiched between a top graphene layer and a bottom graphite thin flake (G/FL-CrI3/Gr). The schematic illustration of our experimental setup with the corresponding optical image of G/FL-CrI3/Gr heterostructure is presented in Fig. 1a, b, respectively. We investigated CrI3 thin flakes with different thickness: from monolayer to few nm in thickness. A representative large-size STM topographic image of G/FL-CrI3/Gr (Fig. 1d) reveals a triangular lattice with a periodicity of $0.69 \pm 0.01$ nm, consistent with the reported lattice constant of CrI3[14]. Therefore, it is very likely that the underlying FL-CrI3 dominates the STM contrast at this particular sample bias $V_s = 1$V, which will be explained in detail later. The intact triangular lattice observed here indicates

structural integrity of the underlying FL-CrI3 flake due to effective protection from the top graphene layer.

### Probing the electronic properties of G/CrI3/Gr.

We then explored local electronic structures of G/FL-CrI3/Gr using scanning tunneling spectroscopy (STS). $dI/dV$ spectrum taken over G/FL-CrI3/Gr (Fig. 2a) reveals two prominent double-peak features above Fermi level ($E_F$), which are labeled as $C_1$ (0.3 V < $V_s$ < 1.1 V) and $C_2$ (1.1 V < $V_s$ < 1.8 V), respectively. A close examination of the $dI/dV$ spectrum in combination with bias-dependent and tunneling current-dependent STM images (Fig. S1 and Fig. S2) allows us to identify the band edges as well as the bandgap of FL-CrI3. We tentatively assign the kink around $V_s = -0.87$ V to the valence band maximum (VBM) of CrI3 and the steep rise at $V_s = 0.26$ V to the conduction band minimum (CBM), which yields a bandgap of 1.13 eV for FL-CrI3, consistent with the reported values obtained by optical measurements[15]. Within the bandgap of FL-CrI3, the $dI/dV$ signal is mainly contributed by graphene as manifested by three characteristic features: (i) a nearly linear DOS as reflected by $dI/dV$ in the sample bias range of $-0.8$ V < $V_s$ < $-0.2$ V, (ii) a gap-like feature around $E_F$ with a sharp increase in $dI/dV$ around $|V_s| = 63$ mV owing to the suppression of the tunneling current due to momentum mismatch and phonon-assisted inelastic tunneling[16], and (iii) a local conduction minimum around $V_S = 0.13$ V associated with the Dirac point of graphene ($E_D$)[16]. For monolayer CrI3 (ML-CrI3), the $dI/dV$ spectrum of G/ML-CrI3/Gr closely resembles that of G/FL-CrI3/Gr (Fig. S3), presumably due to a weak layer dependence of CrI3 electronic structures[17].

### Band structure calculations.

To gain better insight into the electronic structures of G/CrI3, we have performed spin-polarized band structure calculations using the Hubbard-corrected local density approximation (LDA + U). The calculations employed a (5 × 5) supercell of graphene placed on a ($\sqrt{3} \times \sqrt{3}$) supercell of ML-CrI3. The calculated band structure (Fig. 2c) shows that the flat bands of CrI3 hybridize with graphene Dirac cones in the majority-spin channel and electrons transfer from graphene to CrI3, in good agreement with previous density–functional theory (DFT) calculations of G/CrI3[18–20]. A direct comparison of the experimental $dI/dV$ spectra with the theoretical DOS reveals that two double-peak features ($C_1$ and $C_2$) in the $dI/dV$ spectra arise from relatively flat conduction bands of G/CrI3. Specifically, $C_1$ is

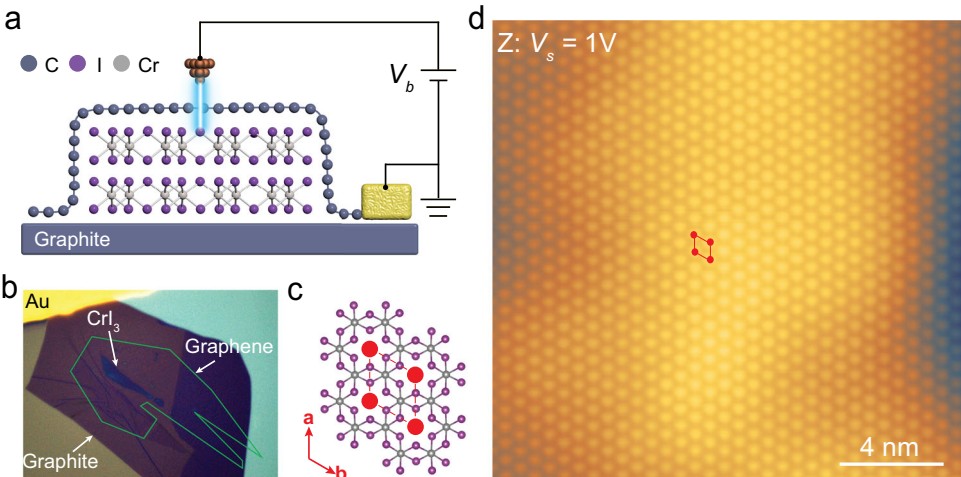

**Fig. 1 The vdW heterostructure of G/FL-CrI3/Gr for STM study. a** The schematic illustration and **b** the optical image of our experimental setup. Our sample consists of monolayer graphene covering FL-CrI3 stacking on graphite flake (G/FL-CrI3/Gr). **c** The atomic structure of monolayer CrI3 (top view). **d** Large-size STM topographic image of G/FL-CrI3/Gr ($V_s = 1$ V, $I_t = 0.1$ nA).

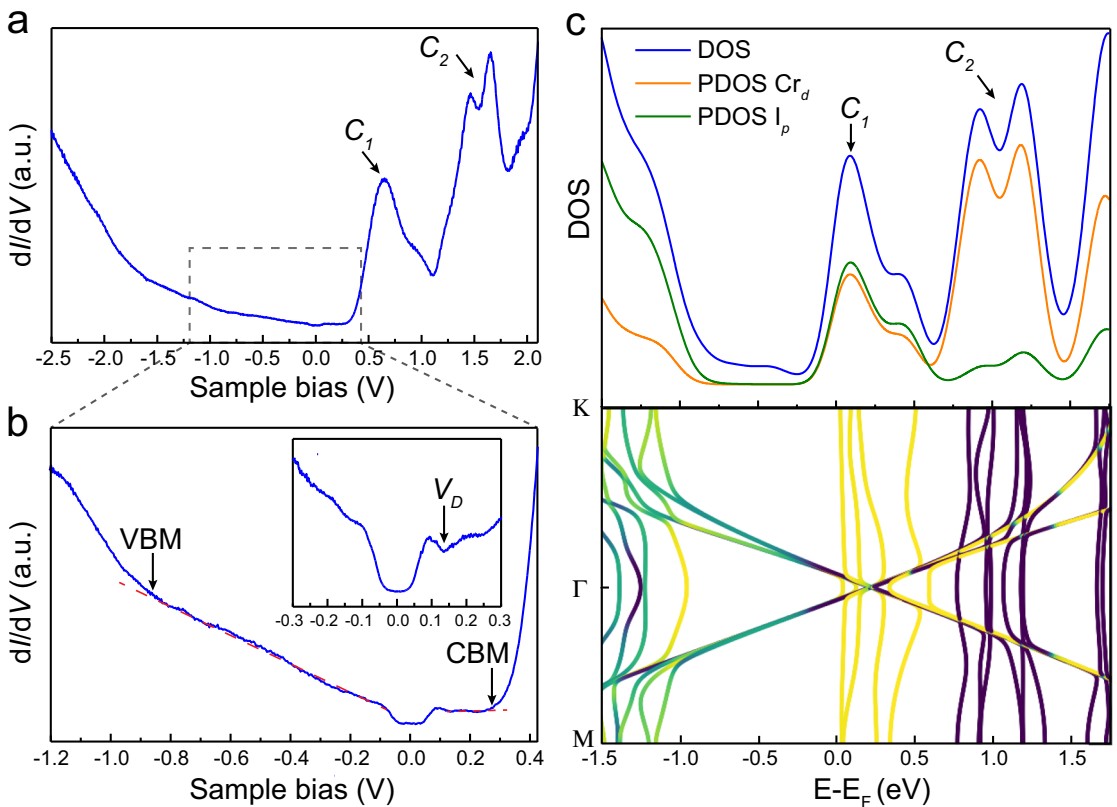

**Fig. 2 The electronic structure of G/CrI$_3$. a** The d$I$/d$V$ spectrum of G/FL-CrI$_3$/Gr taken in a large sample bias window ($-2.5\,V \leq V_s \leq 2.1\,V$). Two prominent double-peak features are indicated by $C_1$ and $C_2$, respectively. **b** The d$I$/d$V$ spectrum of G/FL-CrI$_3$/Gr taken in a small sample bias window ($-1.2\,V \leq V_s \leq 0.42\,V$). The band edges are indicated by VBM and CBM. The inset shows the d$I$/d$V$ spectrum near the Fermi level ($-0.3\,V \leq V_s \leq 0.3\,V$). The local conductance minimum is indicated by $V_D$. **c** Calculated density of states (DOS) and band structure of G/ML-CrI$_3$ using Hubbard $U = 0.5$ eV. Both DOS and the projected DOS (PDOS) on iodine $p$ orbitals and chromium $d$ orbitals are shown. The color-coding in the band structure indicates the expectation value of spin $S_z$ with yellow and purple corresponding to spin-up and -down, respectively.

contributed by majority-spin states ($0\,eV < E - E_F < 0.5\,eV$) with a nearly equal contribution from Cr $d$ states and I $p$ states, both of which strongly hybridize with majority-spin Dirac cones. In contrast, $C_2$ is contributed mainly by minority-spin $d$ states of Cr ($0.8\,eV < E - E_F < 1.3\,eV$) with a negligible hybridization with graphene states.

We note that the Hubbard U has a negligible influence on the overall band shape but significantly changes the energy spacing between $C_1$ and $C_2$. It is found that the use of a Hubbard U of 0.5 eV yields an energy spacing around 0.8 eV between $C_1$ and $C_2$, in good agreement with that observed in the d$I$/d$V$ spectra (Fig. S4). The bandgap of ML-CrI$_3$ predicted from spin-polarized DFT calculations is around 1.24 eV, close to the bandgap measured experimentally. Based on calculated band structures of G/CrI$_3$ (Fig. 2c), it is noted that $E_D$ lies in the bottom of conduction bands of CrI$_3$ in contrast to our experimental observation that $E_D$ is located inside the bandgap. Such a discrepancy is attributed to the additional charge transfer between the bottom graphite substrate and G/CrI$_3$.

We also found that graphene and the underlying CrI$_3$ lattice can be selectively imaged by choosing an appropriate sample bias. Figure 3a–c shows three representative bias-dependent STM images taken on G/FL-CrI$_3$/Gr (a full set of bias-dependent STM images is shown in Fig. S1). The honeycomb lattice of graphene can be clearly resolved at low sample bias ($V_s = -0.3\,V$) within the bandgap (Fig. 3c), while CrI$_3$ lattice with two distinct patterns can be imaged at large sample biases outside the bandgap (Fig. 3a, b). STM image acquired at $V_s = 2.5\,V$ (Fig. 3a) shows a periodic

triangular "cluster" pattern with a lattice constant of $0.69 \pm 0.01$ nm. We then superimposed the atomic structure of ML-CrI$_3$ over the corresponding STM image in Fig. 3a (note that the bottom I atoms in the atomic model are removed for clarity) for a close examination, which reveals that individual triangular clusters are formed by three nearest I atoms in the top atomic plane. In addition, the maxima of triangular cluster protrusion are located at the center of three nearest I atoms in the top atomic plane, equivalent to the center of the hexagon formed by six adjacent Cr atoms. This is similar to the reported STM image of CrBr$_3$[21]. By contrast, the STM image taken at $V_s = -2.5\,V$ (Fig. 3b) shows that the maxima of the protrusion are nearly located over the I atoms in the top atomic plane.

Both STS and STM results indicate that graphene is almost transparent to tunneling electrons when the sample bias is outside the bandgap of FL-CrI$_3$. Otherwise, it would not be possible to probe the atomic structures and electronic properties of the underlying insulating CrI$_3$ flake as semimetallic graphene is closer to the tip by ~3.5 Å (predicted by DFT calculations). Such transparency of graphene in the tunneling process has been observed for graphene grown on metallic substrate[11–13]. It turns out that the substrate states can extend further beyond graphene because graphene's $\pi$ states are strongly localized by both the large in-plane wave vector of graphene's $\pi$ states and the small out-of-plane extension of their atomic orbitals[11]. In the case of G/CrI$_3$ heterostructure, our DFT calculations also confirm that the CrI$_3$ states dominate the simulated STM images at a distance around 4 Å above graphene surface (refer to supporting

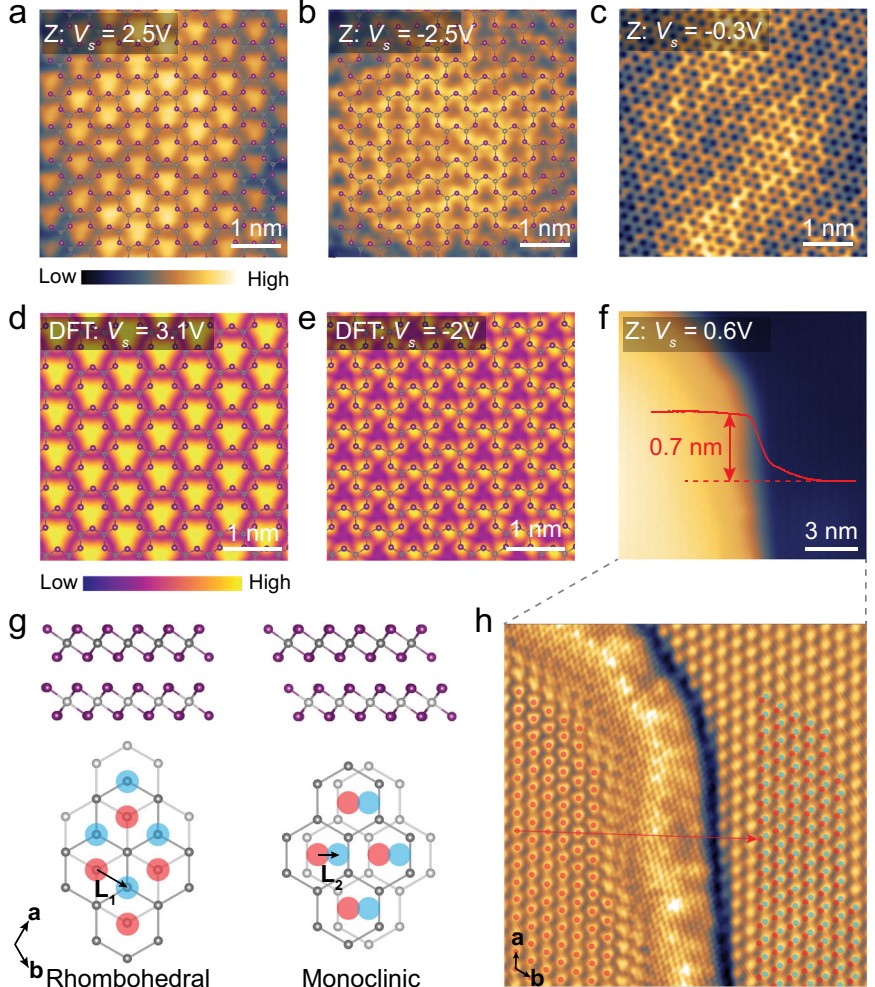

**Fig. 3 STM measurements of G/FL-CrI₃/Gr. a–c** Bias-dependent STM images of G/FL-CrI₃/Gr. STM images taken at (**a**) $V_s = 2.5$ V and (**b**) at $V_s = -2.5$ V with the superimposed atomic structure of ML-CrI₃ (I atoms on the bottom atomic plane are removed for clarity). STM images taken at (**c**) $V_s = -0.3$ V. The tunneling current is $I_t = 1$ nA. **d, e** Simulated STM images taken at (**d**) $V_s = 3.1$ V and (**e**) $V_s = -2$ V with the superimposed atomic structure of ML-CrI₃. **f** STM image acquired across the single-layer step of CrI₃ ($V_s = 0.6$ V, $I_t = 0.2$ nA). **g** The atomic structure of adjacent CrI₃ layers with rhombohedral stacking and monoclinic stacking. The upper (lower) panels are side (top) views. The top view shows the honeycomb lattice formed by Cr atoms (I atoms are removed for clarity), where the center of each hexagon in the upper (lower) layer is indicated by the red (blue) circle. **h** The processed STM image of **f** by using edge enhancement filters to better visualize the atomic lattice of both layers. The lattice of the upper (lower) layer is represented by the red (blue) circle. To intuitively show the atomic translation between two layers, a replica of the upper-layer lattice (translated by ($8a + 16b$) with respect to the original lattice of the upper layer) is shown as the red circle on the lower layer. The red arrow represents the vector ($8a + 16b$).

information S4 for more details). However, the spatial structure of wavefunctions in the regions of space where they have decayed by more than a factor of $10^6$ becomes unreliable (Fig. S5). Therefore, we have chosen to focus on STM simulations at distances of 3 Å above the graphene layer with the introduction of a damping weightage to mimic the structure at larger distances. The simulated STM images at both positive and negative sample biases (Fig. 3d, e) show good agreement with the corresponding experimental STM images (Fig. 3a, b).

**Visualizing the stacking order in the exfoliated FL-CrI₃.** Stacking-dependent interlayer magnetism is another peculiar feature in 2D magnetic insulators. It has been predicted that the monoclinic stacking favors interlayer antiferromagnetic (AFM) coupling, while the rhombohedral stacking favors the interlayer ferromagnetic (FM) coupling[15,17,22–24]. Bulk CrI₃ undergoes a structural phase transition from a monoclinic to a rhombohedral phase at 220 K accompanied with the interlayer FM coupling below the critical temperature of 61 K[14]. By contrast, various

reports have suggested that exfoliated FL-CrI₃ thin flakes show interlayer AFM coupling below the critical temperature[2,6,20,25–29]. This was interpreted as FL-CrI₃ exfoliated at room temperature is kinetically trapped in the monoclinic phase upon cooling (which favors interlayer AFM coupling)[17,22,29]. Such a hypothesis is further verified in recent works by monitoring the change in the second harmonic generation of bilayer CrI₃ during its AFM-to-FM transition[30] and phase-sensitive Raman modes of FL-CrI₃[29]. However, direct atomic-scale visualization of the low-temperature phase of exfoliated FL-CrI₃ is still lacking.

Here, we managed to directly visualize the monoclinic stacking in exfoliated FL-CrI₃ at low temperature by imaging the lateral translation between adjacent CrI₃ layers. Figure 3g illustrates the top and side views of adjacent CrI₃ layers with the rhombohedral and monoclinic stacking. As shown in Fig. 3g, the lower CrI₃ layer is laterally translated by $L_1 = \frac{1}{3}a + \frac{2}{3}b$ ($L_2 = \frac{1}{3}a + \frac{1}{3}b$) with respect to the upper CrI₃ layer for the rhombohedral (monoclinic) stacking[17,29]. The top view shows the honeycomb lattice formed by Cr atoms (I atoms are removed for clarity) for two

different stacking phases, where the center of each hexagon is indicated by the red (blue) circle in the upper (lower) $CrI_3$ layer. As shown in Fig. 3a, the center of each hexagon formed by six adjacent Cr atoms appears as a protrusion in the STM image taken at the positive sample bias. This allows us to identify the lateral translation between adjacent $CrI_3$ layers by examining the STM images of both upper and lower $CrI_3$ layer across a single-layer step. Figure 3f presents a typical STM image of a single-layer step in $CrI_3$ with an expected apparent step height of $0.7 \pm 0.1$ nm[6]. The $CrI_3$ lattice in both upper and lower layers can be better visualized in the STM image processed by the edge enhancement filter in SPIP (Fig. 3h)[31]. We then identify the lattice of both upper and lower layers, which are represented by the red and the blue circles, respectively (Fig. 3h). A statistical analysis shows that the lattice of the lower layer is translated by $L = (8.35 \pm 0.08)a + (16.36 \pm 0.06)b$ with respect to the lattice of the upper layer. Taking the modulus of the translation vector $L$, the lower layer is determined to be translated by $(0.35 \pm 0.08)a + (0.36 \pm 0.06)b$ with respect to the upper layer, which reveals the monoclinic stacking in exfoliated FL-$CrI_3$ at low temperature within the experimental uncertainty. Such a stacking favors the interlayer AFM coupling as predicted by theory[17,22–24].

**Probing the magnetic properties of G/FL-$CrI_3$/Gr.** Apart from the structural and electronic properties of $CrI_3$, we also found that the magnetic properties of underlying FL-$CrI_3$ can be probed through graphene using magnetic field-dependent STM/STS measurements. STM image of G/FL-$CrI_3$/Gr acquired at $V_s = -0.3$ V (Fig. 4a) exhibits the moiré superlattice with a periodicity

of $3.14 \pm 0.01$ nm (refer to the supporting information S5 for more details). The dark (lower) and bright (higher) regions in the topographic STM image are defined as moiré valley and moiré hump, respectively (Fig. 4a). At zero magnetic field, the $dI/dV$ spectra taken in valley and hump regions show a noticeable difference in terms of the energy position and peak intensity around $C_1$ states (Fig. 4e). It is noted that $C_1$ states result from the hybridization of majority-spin $CrI_3$ and graphene states. Therefore, it is very likely that the spatial variation of $C_1$ states in valley and hump regions is originated from the atomic registry-dependent hybridization between graphene and the underlying $CrI_3$[32]. The $dI/dV$ map taken at $V_s = 0.44$ V also captured a spatial moiré modulation of the LDOS (Fig. 4b), consistent with the $dI/dV$ spectroscopic measurement. We then swept the vertical magnetic field and monitored the moiré contrast in the $dI/dV$ maps. As shown in Fig. 4c, d, the characteristic moiré contrast with a nearly constant relative amplitude (defined as the difference in the $dI/dV$ signal between moiré valley and hump as shown in Fig. S7) retains when the magnetic field gradually increases up to 1.84 T. We then ramped the sample bias from 2.2 V to −2.2 V to perform point $dI/dV$ spectroscopy (Fig. S8a). During the measurement, we observed a sudden change of the $I–V$ and $dI/dV$ signal (Fig. S8b). By rescanning the same area at 1.84 T, we found that the characteristic moiré contrast vanished in the $dI/dV$ map ($V_s = 0.44$ V) as shown in Fig. S8c. The magnetic field-dependent moiré contrast in $dI/dV$ maps (taken at fixed bias $V_s = 0.44$ V) is consistent with the evolution of magnetic field-dependent full $dI/dV$ spectra taken over moiré hump and moiré valley (Fig. 4e), which confirms its electronic origin. Upon exposing the sample to 1.84 T, the difference between the

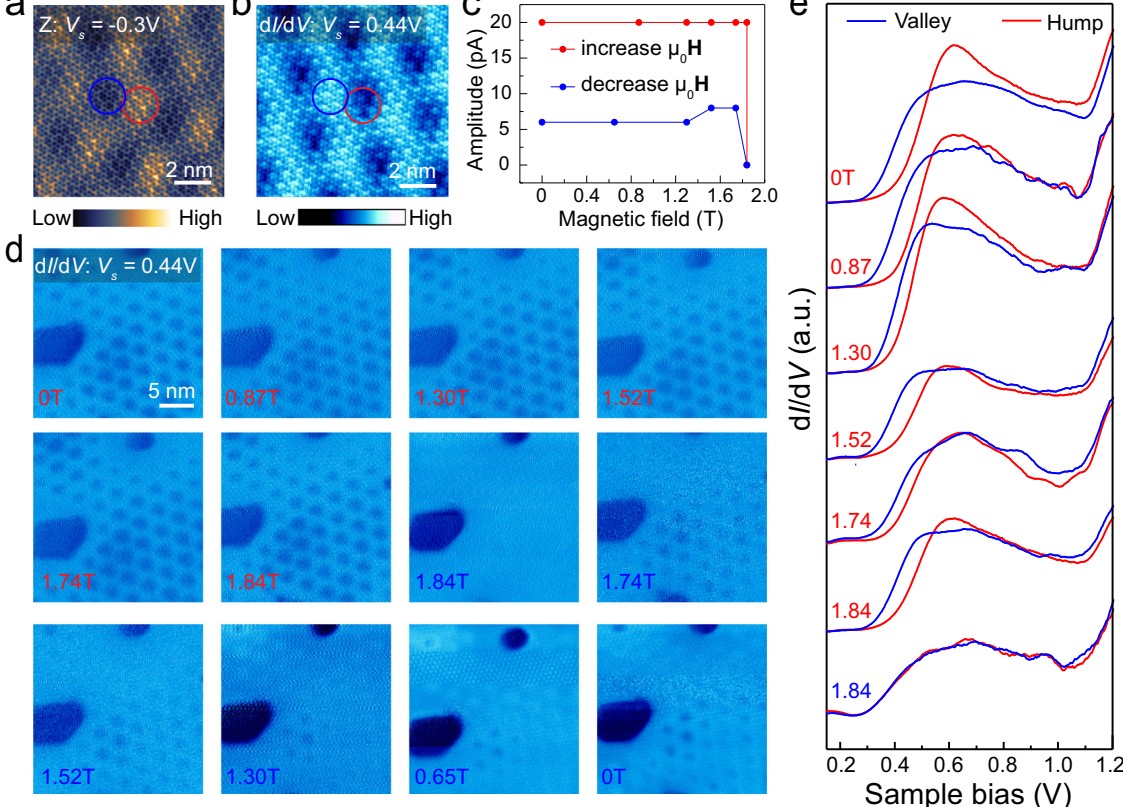

**Fig. 4 Magnetic field-dependent moiré contrast in $dI/dV$ maps. a** STM image ($V_s = -0.3$ V, $I_t = 0.2$ nA) shows the moiré pattern of G/FL-$CrI_3$/Gr. The lower (higher) region is referred as moiré valley (hump) indicated by the blue (red) circle. **b** The corresponding $dI/dV$ map ($V_s = 0.44$ V, $I_t = 0.5$ nA). **c** Magnetic field-dependent moiré contrast in the $dI/dV$ maps ($V_s = 0.44$ V, $I_t = 0.5$ nA). **d** Magnetic field-dependent $dI/dV$ maps ($V_s = 0.44$V, $I_t = 0.5$ nA). **e** Magnetic field-dependent $dI/dV$ spectra taken at moiré valley (blue) and hump (red).

d$I$/d$V$ spectra taken over moiré valley and hump vanishes and they become nearly identical, consistent with the disappearance of the moiré contrast in d$I$/d$V$ maps. Interestingly, as the magnetic field gradually decreases, the moiré contrast reappears but with reduced relative amplitude, resulting in a forward and backward hysteresis (Fig. 4c, d). We note that the critical magnetic field to induce the change of moiré contrasts in different regions of the sample varies from 1.74 to 1.84 T (Figs. S9 and 10), presumably due to the variation of the local environment (like demagnetization field or the formation of domain structures)[6,28].

## Discussion

The magnetic field-dependent moiré contrast in the d$I$/d$V$ maps is likely to be associated with the AFM-to-FM transition in FL-CrI$_3$: the critical magnetic field observed is very close to the typical magnetic field required to align all the spins in different layers of FL-CrI$_3$ (Fig. S11)[2,15,20,29,33,34]. This hypothesis is further corroborated by our spin-polarized DFT calculation of the atomic registry-dependent band structure of G/four-layer CrI$_3$ under AFM and FM interlayer coupling.

Figure 5 shows the calculated band structures of G/four-layer CrI$_3$ (monoclinic phase) with two atomic arrangements (corresponding to the hump and valley) under two magnetic configurations (corresponding to FM and AFM interlayer coupling) (refer to S8 for more details). For the AFM interlayer coupling configurations, only the top CrI$_3$ layer shows a noticeable electronic coupling to graphene in both moiré hump and valley regions, as visualized in Fig. 5a, b. The bands of individual CrI$_3$ layers in AFM-coupled configuration are nearly degenerate and strongly localized in the individual CrI$_3$ layer due to a weak interlayer hybridization (Fig. 5a, b). Because of this, only the bands of the top CrI$_3$ layer hybridize with graphene states and are split off from bands of other CrI$_3$ layers. A careful analysis of the

band structures reveals larger band-splitting energy of the top CrI$_3$ layer in moiré hump compared to moiré valley. This suggests that the electronic hybridization depends strongly on the local atomic registry between graphene and CrI$_3$, which gives rise to a moiré contrast in d$I$/d$V$ maps for G/CrI$_3$ under AFM interlayer coupling.

In contrast, nearly all the four CrI$_3$ layers are electronically coupled to graphene under the FM interlayer coupling configuration. This is because the bands from different CrI$_3$ layers are delocalized over the entire structure (Fig. 5c, d). Although the hybridization still depends on the atomic registry between graphene and individual CrI$_3$ layers, the moiré contrast created by each of the CrI$_3$ layers in graphene is now shifted by a third of the moiré period (due to monoclinic stacking) and thus cancels each other. This explains why the transition to the FM state is seen as the disappearance of the moiré structure.

In conclusion, we have demonstrated a new approach to probe the atomic lattice, intrinsic electronic structure, and interlayer magnetism of mechanically exfoliated FL-CrI$_3$ in a graphene-encapsulated vdW vertical heterostructure using STM/STS. Our results show that overlaid graphene not only protects exfoliated FL-CrI$_3$ from degradation but also allows STM characterization of the underlying FL-CrI$_3$ due to its peculiar transparency to tunneling electrons. The use of semimetallic graphene as a capping layer with electronic transparency to tunneling electrons fulfills the growing demand for the atomic-scale characterization of the artificially assembled vdW heterostructures based on air-sensitive 2D magnetic insulators toward next-generation spintronic devices.

## Methods

**Sample preparation**. The sample is prepared using a well-established dry transfer technique in the glove box. The mechanically exfoliated FL-CrI$_3$ flake is sandwiched between a top graphene layer and a bottom graphite flake. The thickness of

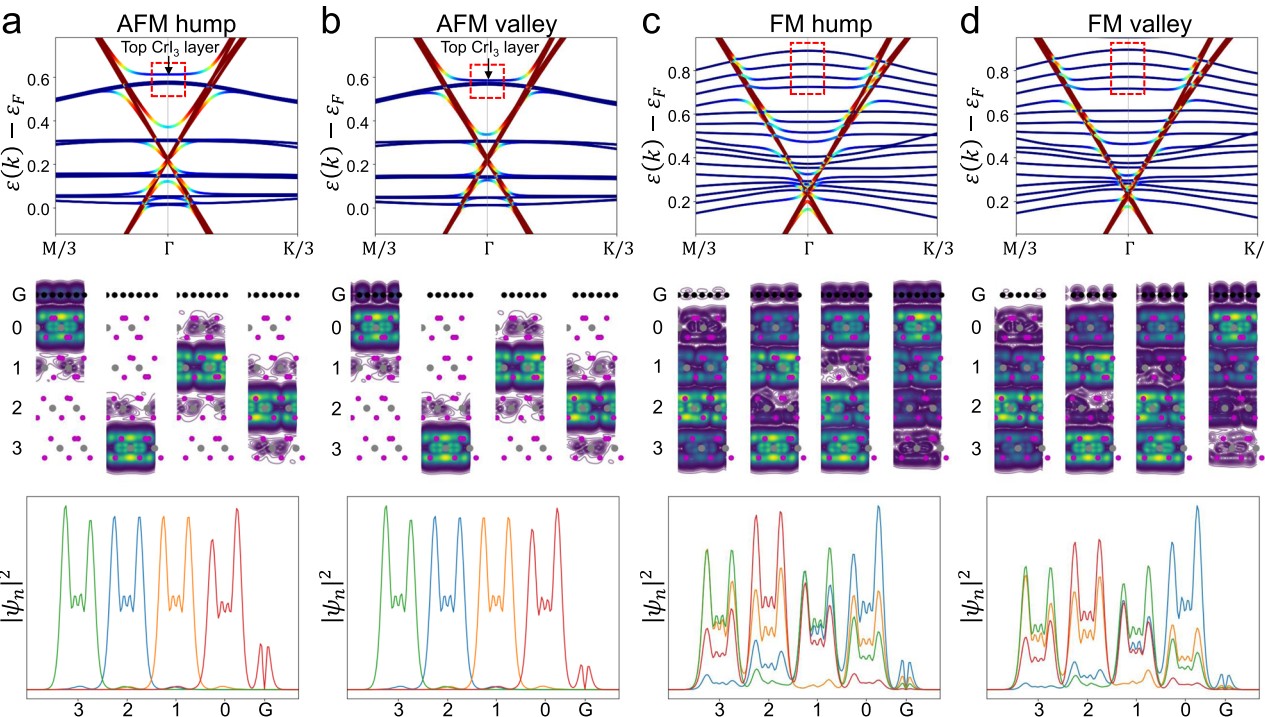

**Fig. 5 The atomic registry-dependent band structure of G/four-layer CrI$_3$ under AFM and FM interlayer coupling. a–d** The band structure and norm-squared wavefunctions of bands in G/four-layer CrI$_3$ with FM and AFM interlayer order in the hump and valley geometries. The top panel shows the band structure of G/four-layer CrI$_3$. We indicate four states using red dashed lines shown in each of the four cases. The middle panel shows contour plots of the norm-squared wavefunctions of indicated states averaged over the $y$-direction of the heterostructure. The bottom panel shows the norm-squared wavefunctions of indicated states averaged over the entire plane.

FL-CrI$_3$ flake is estimated to be ~8–10 nm via the analysis of the corresponding optical contrast. The sample is subjected to UHV annealing at 150 °C for 4 h to ensure the surface cleanness for the STM study.

**STM and STS measurements**. Our STM and STS measurements were conducted at 4.5 K in the Createc LT-STM system with a base pressure lower than 10$^{-10}$ mbar. The tungsten tip was calibrated spectroscopically against the surface state of Au (111) substrate. A superconducting coil was used to apply the out-of-plane external magnetic field with a maximum field of 1.84 T. All the d$I$/d$V$ spectra were measured through a standard lock-in technique with a modulated voltage of 5–10 mV at the frequency of 700–900 Hz.

## Data availability

The data that support the findings of this study are available from the authors on reasonable request, see "Author contributions" for specific data sets.

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

## Acknowledgements

J.L. acknowledges the support from MOE Tier 2 grant (MOE2017-T2-1-056 and R-143-000-A75-114) and NAMIC grant 2019014. M.K. acknowledges support from the Russian Science Foundation (grant # 16-19-10557) and the Ministry of Science and Higher Education of the Russian Federation (0714-2020-0002). K.S.N. also acknowledges support from EU Flagship Programs (Graphene CNECTICT-604391 and 2D-SIPC Quantum Technology), European Research Council Synergy Grant Hetero2D, the Royal Society, and EPSRC grants EP/N010345/1, EP/P026850/1, and EP/S030719/1.

## Author contributions

K.S.N. and J.L. supervised the projects. Z.Q. performed the STM measurements. H.F. and H.Y. helped with the STM measurements. Z.Q., K.S.N., and J.L. analyzed the results. M.H. fabricated the device with contribution from M.K. T.O. performed the theoretical simulation. P.L. and J.L. helped to prepare the sample for the STM study. K.S.N., J.L., and Z.Q. prepared the paper with contribution from T.O. All authors contributed to the scientific discussion and helped in writing the paper.

## Competing interests

The authors declare no competing interests.
