## [Peer Review File · Nature Communications]

Reviewers' Comments:

Reviewer #1:

Remarks to the Author:

In the manuscript, Qiu, et al. present their STM study on graphene covered few-layer CrI₃. The electronic structures of this 2D magnetic insulator can be detected via the tunneling through the top layer graphene. At the magnetic field of 1.84 T, the authors observe the vanishing of the moire patterns of graphene and CrI₃, which is attributed to an AFM to FM transition of CrI₃ under magnetic field. The findings are interesting, but I do have several questions and concerns which the authors need to consider.

1. It is very surprising to see that the moire patterns disappear "slowly" at 1.84 T. As shown in Fig. 4d, the moire patterns are visible at the beginning at 1.84 T, but disappear somehow after "exposing the sample to 1.84 T". What happened here? For me, the time scale of the scanning probe is much longer than that of the magnetic phase transition. If it is really slow, should I expect to see the vanishing of the moire patterns at lower magnetic field with longer exposing time. If this is the case, other explanations (rather than magnetic phase transition) need to be considered.
2. Fig. 3h is clear to show the stacking of the adjacent CrI₃ layers. If the AFM to FM transition occurs, the author should provide similar image to show the structural transition under magnetic field.
3. Fig. 1a is missing.
4. The bias voltages of Fig. 3b are not consistent in the main text and in the figure.

If the authors provide more convincing results to justify their conclusion, I would consider the revised manuscript.

Reviewer #2:

Remarks to the Author:

The manuscript by Qiu et al. reported a STM/STS study of graphene/CrI₃ heterostructure. Recently, the intriguing magnetic property of few-layer CrI₃ has attracted a widespread interest. However, CrI₃ is quite insulating and vulnerable to air exposure, which pose a challenge for STM study. The authors now utilize the semimetallic graphene to cap the few-layer and monolayer CrI₃ prepared by mechanical exfoliation and dry transfer techniques, and thus the atomic structure and electronic property are studied. The authors found that the stacking order in few-layer CrI₃ is monoclinic at low temperature (4.5 K), consistent with previous SHG/Raman measurements and theoretical predictions. The new discovery that the authors made is the magnetic field dependent moire contrast in Fig. 4. By applying an out-of-plane magnetic field, the magnetic state of few-layer CrI₃ could be switched from the layered antiferromagnetism to fully aligned ferromagnetism. The authors found that the hybrid electronic state C1 varies with the magnetic states. While the results are interesting, there remain some questions to be addressed before publication.

1. The magnetic field dependent moire contrast is reported only at the positive magnetic field. The authors shall report the full set of data including the negative magnetic field.
2. Can the authors simultaneously measure the magnetic hysteresis loop by magneto-transport or magneto-optical methods? The combination with the STM data on the same sample will drastically improve the quality and impact of the paper.
3. In figure 3h, the authors analyzed the stacking structure across a single-layer step of CrI₃. Can the author quantify the atomic translation between layers? Also, the error bar is desired.
4. In the caption of figure 4a, the red circle shall be noted as well.

Reviewer #3:

Remarks to the Author:

In this paper by Qiu et al., the authors measured the properties of a few layers exfoliated CrI₃ protected with a graphene monolayer. As a result, they were able to avoid the contamination of CrI₃ and measure at the local scale, using STM/STS, this magnetic insulator thanks to graphene's electronic transparency. Although graphene transparency has been previously used to study its interaction with the underlying substrate, the approach showed in this work opens some interesting possibilities for the STM community to access novel 2D materials.

The authors were able to demonstrate the possibility to image the CrI₃ through the graphene layer and measure the gap with dI/dV curves. They also confirmed the monoclinic stack of the layers and the critical magnetic field for the AFM to FM transition. All these results are in good agreement with previous measurements using different techniques and thus I consider this work a nice proof of principle to study different vdW heterostructures with STM in the future.

In general, the manuscript is well written and the arguments are easy to follow. The manuscript also contains sufficient details for others to be able to perform the experiments again.

There are some minor comments that the authors should consider:

-In line 47 they mention that the "application of vdW technology to the STM will dramatically expand the capabilities" of STM. I will not deny the new opportunities that might come from their results, but I have some doubts about the variety of vdW heterostructures that could be built and measured by STM. I guess that using graphene as a capping layer will always be possible, but I don't know if any other material would allow the access to the underlying material properties. This could be a drawback for the final vdW heterostructures suitable for the experiments.

-From the dI/dV spectra, the authors associate the Dirac point of graphene with the minimum around 0.13V. If this is true, and the graphene is not strongly interacting with the CrI₃, they could try to measure quasi-particle interference to corroborate it (like in ref. 10 from the main text). For that level of doping and if graphene Fermi velocity is not affected, the k vector at Fermi energy should be somewhere around 0.2 nm⁻¹.

-How does the charge transfer affect the band structure of CrI₃? The p doping of graphene is mentioned, but the consequences for the CrI₃ are not clear. A way to study it could be changing the graphene/CrI₃ distance in the calculations and observe the evolution of the bands.

-Did the authors observe any change of visualization from graphene to CrI₃ depending on the current? One would expect the possibility to visualize the graphene layer or the CrI₃ at some voltages just changing the value of the tunneling current.

-Regarding the magnetic field dependent moiré measurements, did the authors measure any other size of moiré? I would expect that for small moirés where the difference in interaction between the lower and higher region is going to be smaller, the contrast difference would be smaller and therefore addressing the value for the AFM-FM transition will be complicated. How much control to obtain different moirés do the authors have?

-Although ref. 10 from the main text was the first one trying to understand the mechanism behind the transparency of graphene, the works on graphene on Au(111) and Ag(111) from the group of Prof. Mikhail Fonin should probably be included (<https://doi.org/10.1038/srep23439> , <https://doi.org/10.1021/nn500396c>).

-In line 29-30 the authors mention the "lack of magnetic sensitivity of traditional STM". I would say that spin polarized STM is a well-established technique now a days and maybe I would not include it in the list with "absence of a conducting path and their extreme air sensitivity".

- The authors should include the tunneling current values in the figure captions.

Point-by-point responses to reviewers of manuscript NCOMMS-20-23819-T

Comments in black - Replies in blue - Amendments to the manuscript in red

Reviewer #1 (Remarks to the Author):

In the manuscript, Qiu, et al. present their STM study on graphene covered few-layer CrI₃. The electronic structures of this 2D magnetic insulator can be detected via the tunneling through the top layer graphene. At the magnetic field of 1.84 T, the authors observe the vanishing of the moiré patterns of graphene and CrI₃, which is attributed to an AFM to FM transition of CrI₃ under magnetic field. The findings are interesting, but I do have several questions and concerns which the authors need to consider.

We thank reviewer for recognizing the novelties of this work, and providing the constructive comments. As described below, we have clarified all the open points indicated by the reviewer. Specific actions are described in the point-by-point responses below.

1. It is very surprising to see that the moiré patterns disappear “slowly” at 1.84 T. As shown in Fig. 4d, the moiré patterns are visible at the beginning at 1.84 T, but disappear somehow after “exposing the sample to 1.84 T”. What happened here? For me, the time scale of the scanning probe is much longer than that of the magnetic phase transition. If it is really slow, should I expect to see the vanishing of the moiré patterns at lower magnetic field with longer exposing time. If this is the case, other explanations (rather than magnetic phase transition) need to be considered.

Thanks for your suggestion. We apologize for a lack of the detailed information about the magnetic phase transition at 1.84 T. We would like to clarify it as follows.

It is observed that moiré pattern (Figure R1a) doesn't disappear immediately upon an increase of magnetic field to 1.84 T, as evidenced in the dI/dV map ($V_s = 0.44V$). We then ramped the sample bias from 2.2 V to -2.2 V to perform the point dI/dV spectroscopy. During the measurement, we observed a sudden change of the $I-V$ and dI/dV signal (Figure R1b). By rescanning the same area at 1.84 T, we found that the characteristic moiré contrast vanished in the dI/dV map ($V_s = 0.44V$), as shown in Figure R1c.

The sudden change of $I-V$ and dI/dV signal is likely to be associated with an abrupt AFM-FM magnetic phase transition induced by the magnetic field with the assistance of tip-induced local gating. This can be attributed to the magneto-electric effect and the influence of doping on magnetization reported in previous studies¹⁻³.

We would like to point out that the critical magnetic field for AFM-FM transition in thin CrI₃ flakes (layer number $n > 2$) is reported to be ranged from 1.60T to 1.92T with an average value of $1.79 \pm 0.10T$ (as shown in Figure S11). The variation of the critical magnetic field is likely due to the sample-to-sample difference, the formation of domain structures and the influence of local environment^{1,4,5}. Unfortunately, 1.84 T is the upper limit of the magnetic

field in our system, which may be insufficient to flip the spin in certain sample regions. In such a case, AFM-FM phase transition could be triggered with the assistance of tip-induced local gating during dI/dV spectroscopic measurement (Figure R1).

In addition, in the certain region of sample, we also observed that moiré contrast vanished in the dI/dV map immediately at a magnetic field of 1.84 T as shown in Figure R2.

Actions:

[1] We have included the sentence below in the revised manuscript (page 8, paragraph 1):

“We then ramped the sample bias from 2.2 V to -2.2 V to perform point dI/dV spectroscopy (Figure S8a). During the measurement, we observed a sudden change of the $I-V$ and dI/dV signal (Figure S8b). By rescanning the same area at 1.84 T, we found that the characteristic moiré contrast vanished in the dI/dV map ($V_s = 0.44V$) as shown in Figure S8c.”

[2] We have included Figure R1 with the description and the sentence below in the revised supplementary information (Figure S8 in S6):

“The sudden change of $I-V$ and dI/dV signal is likely to be associated with an abrupt AFM-FM magnetic phase transition induced by the magnetic field with the assistance of tip-induced local gating. This can be attributed to the magneto-electric effect and the influence of doping on magnetization reported in previous studies⁴⁻⁶.”

[3] We have included Figure R2 with the description in the revised supplementary information (Figure S10 in S6).

[4] We have included the sentence below in the description of Figure S11 in the revised supplementary information:

“The critical magnetic field for AFM-FM transition in thin CrI_3 flakes (layer number $n > 2$) is reported to be ranged from 1.60T to 1.92T (average value: $1.79 \pm 0.10\text{T}$). The variation of the critical magnetic field is likely due to the sample-to-sample difference, the formation of domain structures and the influence of local environment^{4,13,14}. Unfortunately, 1.84 T is the upper limit of the magnetic field in our system, which may be insufficient to flip the spin in certain sample regions. In such a case, AFM-FM phase transition could be triggered with the assistance of tip-induced local gating during dI/dV spectroscopic measurement, as illustrated Figure S8. In addition, in the certain region of sample, we also observed that moiré contrast vanished in the dI/dV map immediately at a magnetic field of 1.84 T as shown in Figure S10.”

Figure R1. AFM-FM magnetic phase transition induced by the magnetic field with the assistance of tip-induced local gating. (a) The dI/dV map ($V_s = 0.44V$, $I_t = 0.5nA$) upon the application of $1.84T$ magnetic field. We then took point dI/dV spectroscopy (the tip position is indicated by the red dot). (b) A sudden change of the I - V and dI/dV signal. (c) The dI/dV map ($V_s = 0.44V$, $I_t = 0.5nA$) taken after the sudden change of the dI/dV signal.

Figure R2. Moiré contrast in the dI/dV map vanished right after applying a magnetic field of $1.84T$ in certain sample regions. (a) Magnetic field dependent dI/dV maps of G/FL- CrI_3 /Gr ($V_s = 0.44V$, $I_t = 1nA$). (b) Magnetic field dependent moiré contrast in the dI/dV maps ($V_s = 0.44V$, $I_t = 1nA$).

2. Fig. 3h is clear to show the stacking of the adjacent CrI_3 layers. If the AFM to FM transition occurs, the author should provide similar image to show the structural transition under magnetic field.

Thanks for your suggestion. An external magnetic field will exert a torque on a magnetic dipole and tend to line up the magnetic moment. As a result, an external magnetic field will lower the potential energy of FM state and drive the AFM-FM transition. However, it is not likely that the structural transition can be induced by external magnetic field. Especially, a

recent magneto-Raman spectroscopy study provides a direct evidence that the monoclinic phase of the exfoliated few-layer CrI₃ remains at B=9T after AFM-FM magnetic phase transition [McCreary A et al. *Nat. Commun.* 2020, 11(1): 1-8.]⁶. Instead, the structural transition can be induced by applying a large pressure as reported in recent studies^{7,8}.

3. Fig. 1a is missing.

We have revised the Figure 1a accordingly.

4. The bias voltages of Fig. 3b are not consistent in the main text and in the figure.

We have corrected this typo in the revised manuscript.

Reviewer #2 (Remarks to the Author):

The manuscript by Qiu et al. reported a STM/STS study of graphene/CrI₃ heterostructure. Recently, the intriguing magnetic property of few-layer CrI₃ has attracted a widespread interest. However, CrI₃ is quite insulating and vulnerable to air exposure, which pose a challenge for STM study. The authors now utilize the semimetallic graphene to cap the few-layer and monolayer CrI₃ prepared by mechanical exfoliation and dry transfer techniques, and thus the atomic structure and electronic property are studied. The authors found that the stacking order in few-layer CrI₃ is monoclinic at low temperature (4.5 K), consistent with previous SHG/Raman measurements and theoretical predictions. The new discovery that the authors made is the magnetic field dependent moiré contrast in Fig. 4. By applying an out-of-plane magnetic field, the magnetic state of few-layer CrI₃ could be switched from the layered antiferromagnetism to fully aligned ferromagnetism. The authors found that the hybrid electronic state C₁ varies with the magnetic states. While the results are interesting, there remain some questions to be addressed before publication.

We thank the reviewer for recognizing the novelty of this manuscript. As described below, we have clarified all the open points indicated by the reviewer.

1. The magnetic field dependent moiré contrast is reported only at the positive magnetic field. The authors shall report the full set of data including the negative magnetic field.

This is a good suggestion. The collection of a full set of data including negative magnetic field is limited by the instrument. Our STM/STS measurements were conducted in Createc LT-STM system with the magnetic coil made by superconducting wire. The magnetic field in the system is at the early stage of the development. The upper limit of downward (negative) magnetic field is only 1.6T, which is insufficient to induce a AFM-FM magnetic phase transition for a few-layer CrI₃ (The critical magnetic field for AFM-FM transition in few-layer CrI₃ is about 1.79 ± 0.10 T according to previous studies as shown in Figure S11). Therefore, we present a complete set of data within upward (positive) magnetic field, wherein the AFM-FM transition can be captured.

2. Can the authors simultaneously measure the magnetic hysteresis loop by magneto-

transport or magneto-optical methods? The combination with the STM data on the same sample will drastically improve the quality and impact of the paper.

Thanks for your suggestion. STM is a local probe for the investigation of atomic-scale structure and properties of this system, while magneto-transport or magneto-optical methods can provide macroscopic-scale properties of this system. These two complementary sets of measurements may not be always necessarily required in the same paper. We feel the macroscopic-scale study of this system using magneto-transport or magneto-optical measurements is beyond the scope of this work. The justifications are provided as follows.

The focus of this manuscript lies in the investigation of atomic structure and magnetism of 2D magnetic insulators *via* tunneling through graphene. By exploiting the transparency of graphene to tunneling electrons, we can probe atomic-scale structure (atomic lattice, interlayer stacking) and electronic properties of few-layer CrI₃ (FL-CrI₃) *via* STM. In addition, AFM-to-FM transition of FL-CrI₃ can be visualized through the magnetic field dependent moiré contrast in the dI/dV maps due to a change of the electronic hybridization between graphene and spin-polarised CrI₃ bands with different interlayer magnetic coupling.

Our results not only provide a general route to probe atomic-scale electronic and magnetic properties of 2D magnetic insulators, but also provide atomic-insights to understand the magnetic properties and AFM-FM transition transitions of few-layer CrI₃ in the previous studies using magneto-transport and magneto-optical methods⁷⁻¹².

3. In figure 3h, the authors analyzed the stacking structure across a single-layer step of CrI₃. Can the author quantify the atomic translation between layers? Also, the error bar is desired.

We thank referee for the valuable suggestion. We quantify the atomic translation between layers using the method reported in the literature [Chen W, Sun Z, Wang Z, et al. *Science*, 2019, **366**(6468): 983-987.]¹³. We first identified the lattices of both upper layer and lower layer, which are represented by red circles and blue circles, respectively (Figure R3b). A statistical analysis shows that the lattice of the lower layer is translated by $\mathbf{L} = (8.35 \pm 0.08)\mathbf{a} + (16.36 \pm 0.06)\mathbf{b}$ with respect to the lattice of the upper layer. Taking the modulus of the translation vector \mathbf{L} , the lower layer is determined to be translated by $(0.35 \pm 0.08)\mathbf{a} + (0.36 \pm 0.06)\mathbf{b}$ with respect to the upper layer, which reveals the monoclinic stacking in exfoliated few-layer CrI₃ at low temperature within the experimental uncertainty.

Actions:

[1] We have included the sentences below in the revised manuscript (page 7, paragraph 1):

“We then identified the lattices of both upper layer and lower layer, which are represented by the red circle and the blue circle, respectively (Figure 3h). A statistical analysis shows the lattice of the lower layer is translated by $\mathbf{L} = (8.35 \pm 0.08)\mathbf{a} + (16.36 \pm 0.06)\mathbf{b}$ with respect to the lattice of the upper layer. Taking the modulus of the translation vector \mathbf{L} , the lower layer is determined to be translated by $(0.35 \pm 0.08)\mathbf{a} + (0.36 \pm 0.06)\mathbf{b}$ with respect to the upper layer, which reveals the monoclinic stacking in exfoliated FL-CrI₃ at low

temperature within the experimental uncertainty. Such a stacking favors the interlayer AFM coupling as predicted by theory^{7,20,24,25}."

[2] We have replaced Figure 3h with Figure R3b and revised description in the main text.

Figure R3. Quantify the atomic translation between layers. (a) The atomic structure of adjacent CrI₃ layers with rhombohedral stacking and monoclinic stacking. The upper (lower) panels are side (top) views. The top view shows the honeycomb lattice formed by Cr atoms (I atoms are removed for clarity), where the center of each hexagon in the upper (lower) layer is indicated by the red (blue) circles. (b) The processed STM image by using edge enhancement filters to better visualize the atomic lattice of both layers. The lattice of the upper (lower) layer are represented by the red (blue) circle. To intuitively show the atomic translation between two layers, a replica of the upper layer lattice (translated by $(8a + 16b)$ with respect to the original lattice of the upper layer) is shown as the red circle on the lower layer. The red arrow represents the vector $(8a + 16b)$.

4. In the caption of figure 4a, the red circle shall be noted as well.

Thanks for your suggestion. We have corrected it in our revised manuscript.

Reviewer #3 (Remarks to the Author):

In this paper by Qiu et al., the authors measured the properties of a few layers exfoliated CrI₃ protected with a graphene monolayer. As a result, they were able to avoid the contamination of CrI₃ and measure at the local scale, using STM/STS, this magnetic insulator thanks to graphene's electronic transparency. Although graphene transparency has been previously used to study its interaction with the underlying substrate, the approach showed in this work

opens some interesting possibilities for the STM community to access novel 2D materials. The authors were able to demonstrate the possibility to image the CrI₃ through the graphene layer and measure the gap with dI/dV curves. They also confirmed the monoclinic stack of the layers and the critical magnetic field for the AFM to FM transition. All these results are in good agreement with previous measurements using different techniques and thus I consider this work a nice proof of principle to study different vdW heterostructures with STM in the future.

In general, the manuscript is well written and the arguments are easy to follow. The manuscript also contains sufficient details for others to be able to perform the experiments again.

We thank the reviewer for pointing out that our work offers new possibility for STM to access novel 2D materials, and for providing the constructive suggestions for further improvement. As described below, we have also carefully revised our manuscript and clarified all the points suggested by the reviewer

There are some minor comments that the authors should consider:

-In line 47 they mention that the "application of vdW technology to the STM will dramatically expand the capabilities" of STM. I will not deny the new opportunities that might come from their results, but I have some doubts about the variety of vdW heterostructures that could be built and measured by STM. I guess that using graphene as a capping layer will always be possible, but I don't know if any other material would allow the access to the underlying material properties. This could be a drawback for the final vdW heterostructures suitable for the experiments.

We agree with referee that application of vdW technology to the STM will provide new opportunities for the study of vdW heterostructures.

Such a transparency of graphene in the tunneling process has also been observed for graphene grown on metallic substrate¹⁴. It turns out that the substrate states can extend further beyond the graphene because the graphene's π states are strongly localized by both the large in-plane wave vector of graphene's π states and the small out-of-plane extension of their atomic orbitals¹⁴. We think this is a general picture that can be utilized to probe vdW heterostructures.

In the case of G/CrI₃ heterostructure, our DFT calculations also confirm that the CrI₃ states dominate the simulated STM images at a distance around 4 Å above graphene surface.

Apart from G/2D magnetic insulators, we also performed STM/STS study of a different vdW heterostructure consisting of graphene on a semiconducting monolayer ReSe₂ (G/ML-ReSe₂) (Figure R4a) in our previous work [Qiu, Zhizhan, et al. *Sci. Adv.* **5.7** (2019): eaaw2347.]¹⁵. In this work, we unambiguously resolved the conduction band, valence band and bandgap of the ML-ReSe₂ under graphene (blue curve in Figure 4c). We further reversed the stacking order to fabricate a ML-ReSe₂/G heterostructure (Figure R4b). STS measurements reveal that the electronic band structure of ML-ReSe₂ in these two different heterostructures (G/ML-ReSe₂ and ML-ReSe₂/G) are nearly identical (Figure R4c). Very recently, we also performed STM/STS study of graphene on a new 2D material, which proves the same concept

(unpublished results). All these observations point out that such a concept (the application of vdW technology to the STM) has the general validity.

Therefore, we believe that this method provides an attractive platform to explore different types of vdW heterostructures *via* exploiting the transparency of graphene to tunneling electrons.

Action:

We have included the reference [Qiu, Zhizhan, et al. *Sci. Adv.* **5.7** (2019): eaaw2347.] as ref. 16 in the main text. We also include references [Tesch, J. et al. *Sci. Rep.* **6**, 23439 (2016); Leicht, P. et al. *ACS nano* **8**, 3735-3742 (2014).] as ref. 11-12 in the main text.

Figure R4. Probing the electronic properties of ML-ReSe₂ in graphene based vdW heterostructures. (a) The schematic illustration of a back-gated G/ML-ReSe₂ device. (b) The schematic illustration of a back-gated ML-ReSe₂/G device. (c) The dI/dV spectra on G/ML-ReSe₂ (blue) and ML-ReSe₂/G (red).

-From the dI/dV spectra, the authors associate the Dirac point of graphene with the minimum around 0.13V. If this is true, and the graphene is not strongly interacting with the CrI₃, they could try to measure quasi-particle interference to corroborate it (like in ref. 10 from the main text). For that level of doping and if graphene Fermi velocity is not affected, the k vector at Fermi energy should be somewhere around 0.2 nm⁻¹.

Thanks for your suggestion. Quasi-particle interference (QPI) occurs when there is a scatter center (such as defects) that can scatter the electron from one state to another. In view of referee's suggestion, we attempted to measure QPI over a surface region that contains various defects (Figure R5a). Unfortunately, the QPI observed is so weak, resulting in a non-discernible feature in both dI/dV maps (Figure R5b-d) and corresponding FFT images (Figure

R5f-h). We note the system reported in previous studies of QPI in graphene usually contains various defects with atomically-sharp contrast^{14,16}. In contrast, defects in G/FL-CrI₃ show a slowly varying impurity potential, much larger than the graphene lattice constant (Figure R5b-d). Such a long-range or medium-range perturbation can be neglected as scattering mechanisms [Ando T, Nakanishi T. *J. Phys. Soc. Jpn.*, 1998, **67**(5): 1704-1713.]¹⁷. Therefore, the energy position of Dirac point cannot be inferred by QPI measurement, unlike the case in ref [González-Herrero, H. c. et al. *ACS Nano* **10**, 5131-5144 (2016).]. A further study of the energy position of Dirac point is an interesting topic. However, this requires the use of external gate as a tuning knob or through the transport measurement, which is outside the scope of this paper.

Figure R5. The weak QPI in G/FL-CrI₃. (a) The STM image ($V_s = 0.63 V, I = 1 nA$) showing lots of defects. Defects are outlined by black circles. (b-d) dI/dV maps taken at different sample bias: $V_s = 0.63 V$ (b), $V_s = 0.44 V$ (c) and $V_s = 0.25 V$ (d). The set point of tunneling current is $I_t = 1 nA$. (e) QPI in the first Brillouin zone (BZ) of graphene. The black dot is the reciprocal lattice of graphene. q_F is the Fermi wave vector. The BZ of graphene is indicated by the black dashed hexagon. The ring of diameter $4q_F$ at Γ (red) is related to the intravalley scattering process and the ring of diameter $4q_F$ at K and K' (green) is related to the intervalley scattering process. (f-g) The FFT images of dI/dV maps shown in panel (b-d). The graphene reciprocal lattice is outlined by the white circle.

-How does the charge transfer affect the band structure of CrI₃? The p doping of graphene is mentioned, but the consequences for the CrI₃ are not clear. A way to study it could be changing the graphene/CrI₃ distance in the calculations and observe the evolution of the bands.

Thanks for your suggestion. A direct comparison between the calculated band structure of ML-CrI₃ and G/ML-CrI₃ reveals that the charge transfer will shift the Fermi level into the bottom of CrI₃ conduction band but it has negligible influence on the overall band shape of

CrI₃ [Zhang, J. et al. *Phys. Rev. B* **97**, 085401 (2018)]¹⁸ (this reference has been cited as ref. 22 in the main text).

As discussed in the main text (page 5, paragraph 1), there is some discrepancy between predicted band structures with the respect to the Fermi level and experimental data, which can be attributed to the following two factors: (i) A suspending G/CrI₃ model (without substrate) is calculated in the DFT, while G/CrI₃ sample is placed on a graphite substrate. There may be charge transfer between G/CrI₃ and graphite substrate as mentioned in the main text. (ii) it is likely that the presence of defects in CrI₃ also introduce the doping and change the overall Fermi level.

The impact of the interface distance on G/CrI₃ bands has been explored in the previous work [Zhang, J. et al. *Phys. Rev. B* **97**, 085401 (2018)]¹⁸. Our results show good agreement with the calculated band structure of G/CrI₃ at equilibrium interface distance in this Ref [Zhang, J. et al. *Phys. Rev. B* **97**, 085401 (2018)]. In addition, the authors also found that as a reduction of interface distance, the FM CrI₃ substrate will induce a very large magnetic exchange field and Rashba SOC in graphene, which may potentially lead to Chern insulating state. In addition, a decrease of the interface distance will move the Fermi level from the position near the bottom of the conduction bands to the top of the valence bands, indicating the variation from Ohmic contacts to Schottky contacts at the G/CrI₃ interface. The theoretical results reported by Zhang, J. et al. point out a potential route to realize Chern insulating state in graphene by compressing graphene/CrI₃. Such a calculation is not closely related to our experimental results.

-Did the authors observe any change of visualization from graphene to CrI₃ depending on the current? One would expect the possibility to visualize the graphene layer or the CrI₃ at some voltages just changing the value of the tunneling current.

Thanks for your comment. It is noted that the visualization of graphene or CrI₃ lattice mainly depends on the sample bias rather than tunnelling current. Figure R6a-d shows the STM images taken at $V_s = 1V$ by varying the tunneling current from 0.1 nA to 1 nA. All these STM images resolves CrI₃ lattice, independent of tunnelling current. That is because the electronic states probed are deep into the CrI₃ conduction band and thus the CrI₃ states have a major contribution to the tunneling current.

A smaller positive bias ($V_s = 0.44V$) probes the electronic states near the CrI₃ conduction band edge as well as graphene states. In this case, the weight of integrated graphene states and integrated CrI₃ states may be comparable so that both graphene lattice and CrI₃ lattice can be resolved (Figure R5e-h). A variation of tunneling current from 0.1 nA to 1 nA has a limited influence on the visualization of graphene lattice. An increase of tunneling current makes the graphene (CrI₃) lattice slightly more (less) prominent.

Action:

[1] We have included Figure R6 with the description in the revised supplementary information to present STM images recorded at varied tunnelling currents (sample bias is kept the same for each set of data) (Figure S2 in S1).

Figure R6. STM images of G/FL-CrI₃/Gr recorded at varied tunneling current. (a-d) STM images ($V_s = 1V$) recorded at varied tunneling current from 0.1 nA to 1 nA. Note the STM images shown could be taken on different areas. (e-h) STM images ($V_s = 0.44V$) recorded at varied tunneling current from 0.1 nA to 1 nA.

-Regarding the magnetic field dependent moiré measurements, did the authors measure any other size of moiré? I would expect that for small moirés where the difference in interaction between the lower and higher region is going to be smaller, the contrast difference would be smaller and therefore addressing the value for the AFM-FM transition will be complicated. How much control to obtain different moirés do the authors have?

We agree with referee that moiré modulation is likely to be weaker for a smaller moiré superlattice, as suggested by previous studies in other moiré systems^{19,20}. We have performed STM/STS study of a moiré superlattice with a periodicity of 1.84 nm (Figure R7a). Both spatially dependent dI/dV spectra (Figure R7c) and dI/dV map (Figure R7b) show a non-discernible spatial moiré modulation on the LDOS at zero magnetic field. Therefore, a negligible moiré modulation in a small moiré superlattice makes it impossible to address AFM-FM magnetic phase transition via the visualization of magnetic field dependent moiré contrast in dI/dV maps.

Figure R7. The non-discernible moiré modulation in a small moiré superlattice. (a) a small moiré superlattice with a periodicity of 1.84 nm as shown in the STM image ($V_s =$

0.4V, $I_t = 0.5nA$). The unit cell of moiré superlattice is outlined by the black curve. (b) The corresponding dI/dV map ($V_s = 0.4V, I_t = 0.5nA$). (c) Spatially dependent dI/dV spectra taken at different locations. The tip position where we took the blue (red) dI/dV spectrum is indicated by the blue (red) cross in panel (a-b).

-Although ref. 10 from the main text was the first one trying to understand the mechanism behind the transparency of graphene, the works on graphene on Au(111) and Ag(111) from the group of Prof. Mikhail Fonin should probably be included

(<https://doi.org/10.1038/srep23439> , <https://doi.org/10.1021/mn500396c>).

Thanks for your suggestion. We have included the references [Tesch, J. *et al. Sci. Rep.* **6**, 23439 (2016); Leicht, P. *et al. ACS Nano.* **8**, 3735-3742 (2014).]^{21,22} as ref.11-12 in the revised manuscript.

-In line 29-30 the authors mention the "lack of magnetic sensitivity of traditional STM". I would say that spin polarized STM is a well-established technique nowadays and maybe I would not include it in the list with "absence of a conducting path and their extreme air sensitivity".

We agree with referee. We have revised it accordingly.

Action: We have revised the sentence in the manuscript (page 2, paragraph 1):

“Further progress hinges on deep understanding of electronic and magnetic properties of 2D magnets at the atomic scale. Although local electronic properties can be probed by scanning tunneling microscopy/spectroscopy (STM/STS), its application to investigate 2D magnetic insulators remains elusive due to absence of a conducting path and their extreme air sensitivity¹⁻³.”

- The authors should include the tunneling current values in the figure captions.

Thanks for your suggestion. We have included the tunneling current values in the figure captions in the revised manuscript.

Reference:

- 1 Jiang, S., Shan, J. & Mak, K. F. Electric-field switching of two-dimensional van der Waals magnets. *Nat. Mater.* **17**, 406-410 (2018).
- 2 Jiang, S., Li, L., Wang, Z., Mak, K. F. & Shan, J. Controlling magnetism in 2D CrI₃ by electrostatic doping. *Nat. Nanotech.* **13**, 549-553 (2018).
- 3 Jiang, S., Li, L., Wang, Z., Shan, J. & Mak, K. F. Spin tunnel field-effect transistors based on two-dimensional van der Waals heterostructures. *Nat. Electron.* **2**, 159-163 (2019).
- 4 Huang, B. et al. Layer-dependent ferromagnetism in a van der Waals crystal down to the monolayer limit. *Nature* **546**, 270-273 (2017).

- 5 Thiel, L. et al. Probing magnetism in 2D materials at the nanoscale with single-spin
microscopy. *Science* **364**, 973-976 (2019).
- 6 McCreary, A. et al. Distinct magneto-Raman signatures of spin-flip phase transitions
in CrI₃. *Nat. Commun.* **11**, 1-8 (2020).
- 7 Li, T. et al. Pressure-controlled interlayer magnetism in atomically thin CrI₃. *Nat.*
Mater. **18**, 1303-1308 (2019).
- 8 Song, T. et al. Switching 2D magnetic states via pressure tuning of layer stacking. *Nat.*
Mater. **18**, 1-5 (2019).
- 9 Song, T. et al. Giant tunneling magnetoresistance in spin-filter van der Waals
heterostructures. *Science* **360**, 1214-1218 (2018).
- 10 Klein, D. R. et al. Probing magnetism in 2D van der Waals crystalline insulators via
electron tunneling. *Science* **360**, 1218-1222 (2018).
- 11 Zhong, D. et al. Van der Waals engineering of ferromagnetic semiconductor
heterostructures for spin and valleytronics. *Sci. Adv.* **3**, e1603113 (2017).
- 12 Wang, Z. et al. Very large tunneling magnetoresistance in layered magnetic
semiconductor CrI₃. *Nat. Commun.* **9**, 1-8 (2018).
- 13 Chen, W. et al. Direct observation of van der Waals stacking-dependent interlayer
magnetism. *Science* **366**, 983-987 (2019).
- 14 González-Herrero, H. c. et al. Graphene tunable transparency to tunneling electrons: a
direct tool to measure the local coupling. *ACS Nano* **10**, 5131-5144 (2016).
- 15 Qiu, Z. et al. Giant gate-tunable bandgap renormalization and excitonic effects in a
2D semiconductor. *Sci. Adv.* **5**, eaaw2347 (2019).
- 16 Rutter, G. M. et al. Scattering and interference in epitaxial graphene. *Science* **317**,
219-222 (2007).
- 17 Ando, T. & Nakanishi, T. Impurity scattering in carbon nanotubes—absence of back
scattering. *J. Phy. Soc. Jpn.* **67**, 1704-1713 (1998).
- 18 Zhang, J. et al. Strong magnetization and Chern insulators in compressed
graphene/CrI₃ van der Waals heterostructures. *Phys. Rev. B* **97**, 085401 (2018).
- 19 Decker, R. et al. Local electronic properties of graphene on a BN substrate via
scanning tunneling microscopy. *Nano Lett.* **11**, 2291-2295 (2011).
- 20 Alexeev, E. M. et al. Resonantly hybridized excitons in moiré superlattices in van der
Waals heterostructures. *Nature* **567**, 81-86 (2019).
- 21 Tesch, J. et al. Structural and electronic properties of graphene nanoflakes on Au (111)
and Ag (111). *Sci. Rep.* **6**, 23439 (2016).
- 22 Leicht, P. et al. In situ fabrication of quasi-free-standing epitaxial graphene
nanoflakes on gold. *ACS Nano* **8**, 3735-3742 (2014).

Reviewers' Comments:

Reviewer #1:

Remarks to the Author:

The authors have addressed my concerns and I would recommend to publish the revised manuscript in Nature Communications.

Reviewer #2:

Remarks to the Author:

I have carefully read through the response letter and the revised manuscript. The authors have addressed most of the questions/comments raised by me and other referees. I just feel it is a pity that the authors' setup cannot allow them to sweep down the magnetic field far enough to obtain the full set of data. I believe this kind of data could help further understand the vanishing of moire pattern across the AFM-FM phase transition.

Reviewer #3:

Remarks to the Author:

I am satisfied with the response of the authors to all my questions and the changes they made in the main and supporting text. I have no objection against publication of the manuscript in its present form.

Point-by-point responses to reviewers of manuscript NCOMMS-20-23819-T

Comments in black - Replies in blue - Amendments to the manuscript in red

Reviewer #1 (Remarks to the Author):

In the manuscript, Qiu, et al. present their STM study on graphene covered few-layer CrI₃. The electronic structures of this 2D magnetic insulator can be detected via the tunneling through the top layer graphene. At the magnetic field of 1.84 T, the authors observe the vanishing of the moiré patterns of graphene and CrI₃, which is attributed to an AFM to FM transition of CrI₃ under magnetic field. The findings are interesting, but I do have several questions and concerns which the authors need to consider.

We thank reviewer for recognizing the novelties of this work, and providing the constructive comments. As described below, we have clarified all the open points indicated by the reviewer. Specific actions are described in the point-by-point responses below.

1. It is very surprising to see that the moiré patterns disappear “slowly” at 1.84 T. As shown in Fig. 4d, the moiré patterns are visible at the beginning at 1.84 T, but disappear somehow after “exposing the sample to 1.84 T”. What happened here? For me, the time scale of the scanning probe is much longer than that of the magnetic phase transition. If it is really slow, should I expect to see the vanishing of the moiré patterns at lower magnetic field with longer exposing time. If this is the case, other explanations (rather than magnetic phase transition) need to be considered.

Thanks for your suggestion. We apologize for a lack of the detailed information about the magnetic phase transition at 1.84 T. We would like to clarify it as follows.

It is observed that moiré pattern (Figure R1a) doesn't disappear immediately upon an increase of magnetic field to 1.84 T, as evidenced in the dI/dV map ($V_s = 0.44V$). We then ramped the sample bias from 2.2 V to -2.2 V to perform the point dI/dV spectroscopy. During the measurement, we observed a sudden change of the $I-V$ and dI/dV signal (Figure R1b). By rescanning the same area at 1.84 T, we found that the characteristic moiré contrast vanished in the dI/dV map ($V_s = 0.44V$), as shown in Figure R1c.

The sudden change of $I-V$ and dI/dV signal is likely to be associated with an abrupt AFM-FM magnetic phase transition induced by the magnetic field with the assistance of tip-induced local gating. This can be attributed to the magneto-electric effect and the influence of doping on magnetization reported in previous studies¹⁻³.

We would like to point out that the critical magnetic field for AFM-FM transition in thin CrI₃ flakes (layer number $n > 2$) is reported to be ranged from 1.60T to 1.92T with an average value of $1.79 \pm 0.10T$ (as shown in Figure S11). The variation of the critical magnetic field is likely due to the sample-to-sample difference, the formation of domain structures and the influence of local environment^{1,4,5}. Unfortunately, 1.84 T is the upper limit of the magnetic

field in our system, which may be insufficient to flip the spin in certain sample regions. In such a case, AFM-FM phase transition could be triggered with the assistance of tip-induced local gating during dI/dV spectroscopic measurement (Figure R1).

In addition, in the certain region of sample, we also observed that moiré contrast vanished in the dI/dV map immediately at a magnetic field of 1.84 T as shown in Figure R2.

Actions:

[1] We have included the sentence below in the revised manuscript (page 8, paragraph 1):

“We then ramped the sample bias from 2.2 V to -2.2 V to perform point dI/dV spectroscopy (Figure S8a). During the measurement, we observed a sudden change of the $I-V$ and dI/dV signal (Figure S8b). By rescanning the same area at 1.84 T, we found that the characteristic moiré contrast vanished in the dI/dV map ($V_s = 0.44V$) as shown in Figure S8c.”

[2] We have included Figure R1 with the description and the sentence below in the revised supplementary information (Figure S8 in S6):

“The sudden change of $I-V$ and dI/dV signal is likely to be associated with an abrupt AFM-FM magnetic phase transition induced by the magnetic field with the assistance of tip-induced local gating. This can be attributed to the magneto-electric effect and the influence of doping on magnetization reported in previous studies⁴⁻⁶.”

[3] We have included Figure R2 with the description in the revised supplementary information (Figure S10 in S6).

[4] We have included the sentence below in the description of Figure S11 in the revised supplementary information:

“The critical magnetic field for AFM-FM transition in thin CrI_3 flakes (layer number $n > 2$) is reported to be ranged from 1.60T to 1.92T (average value: $1.79 \pm 0.10\text{T}$). The variation of the critical magnetic field is likely due to the sample-to-sample difference, the formation of domain structures and the influence of local environment^{4,13,14}. Unfortunately, 1.84 T is the upper limit of the magnetic field in our system, which may be insufficient to flip the spin in certain sample regions. In such a case, AFM-FM phase transition could be triggered with the assistance of tip-induced local gating during dI/dV spectroscopic measurement, as illustrated Figure S8. In addition, in the certain region of sample, we also observed that moiré contrast vanished in the dI/dV map immediately at a magnetic field of 1.84 T as shown in Figure S10.”

Figure R1. AFM-FM magnetic phase transition induced by the magnetic field with the assistance of tip-induced local gating. (a) The dI/dV map ($V_s = 0.44\text{ V}$, $I_t = 0.5\text{ nA}$) upon the application of 1.84 T magnetic field. We then took point dI/dV spectroscopy (the tip position is indicated by the red dot). (b) A sudden change of the I - V and dI/dV signal. (c) The dI/dV map ($V_s = 0.44\text{ V}$, $I_t = 0.5\text{ nA}$) taken after the sudden change of the dI/dV signal.

Figure R2. Moiré contrast in the dI/dV map vanished right after applying a magnetic field of 1.84 T in certain sample regions. (a) Magnetic field dependent dI/dV maps of G/FL- CrI_3/Gr ($V_s = 0.44\text{ V}$, $I_t = 1\text{ nA}$). (b) Magnetic field dependent moiré contrast in the dI/dV maps ($V_s = 0.44\text{ V}$, $I_t = 1\text{ nA}$).

2. Fig. 3h is clear to show the stacking of the adjacent CrI_3 layers. If the AFM to FM transition occurs, the author should provide similar image to show the structural transition under magnetic field.

Thanks for your suggestion. An external magnetic field will exert a torque on a magnetic dipole and tend to line up the magnetic moment. As a result, an external magnetic field will lower the potential energy of FM state and drive the AFM-FM transition. However, it is not likely that the structural transition can be induced by external magnetic field. Especially, a

recent magneto-Raman spectroscopy study provides a direct evidence that the monoclinic phase of the exfoliated few-layer CrI₃ remains at B=9T after AFM-FM magnetic phase transition [McCreary A et al. *Nat. Commun.* 2020, 11(1): 1-8.]⁶. Instead, the structural transition can be induced by applying a large pressure as reported in recent studies^{7,8}.

3. Fig. 1a is missing.

We have revised the Figure 1a accordingly.

4. The bias voltages of Fig. 3b are not consistent in the main text and in the figure.

We have corrected this typo in the revised manuscript.

Reviewer #2 (Remarks to the Author):

The manuscript by Qiu et al. reported a STM/STS study of graphene/CrI₃ heterostructure. Recently, the intriguing magnetic property of few-layer CrI₃ has attracted a widespread interest. However, CrI₃ is quite insulating and vulnerable to air exposure, which pose a challenge for STM study. The authors now utilize the semimetallic graphene to cap the few-layer and monolayer CrI₃ prepared by mechanical exfoliation and dry transfer techniques, and thus the atomic structure and electronic property are studied. The authors found that the stacking order in few-layer CrI₃ is monoclinic at low temperature (4.5 K), consistent with previous SHG/Raman measurements and theoretical predictions. The new discovery that the authors made is the magnetic field dependent moiré contrast in Fig. 4. By applying an out-of-plane magnetic field, the magnetic state of few-layer CrI₃ could be switched from the layered antiferromagnetism to fully aligned ferromagnetism. The authors found that the hybrid electronic state C₁ varies with the magnetic states. While the results are interesting, there remain some questions to be addressed before publication.

We thank the reviewer for recognizing the novelty of this manuscript. As described below, we have clarified all the open points indicated by the reviewer.

1. The magnetic field dependent moiré contrast is reported only at the positive magnetic field. The authors shall report the full set of data including the negative magnetic field.

This is a good suggestion. The collection of a full set of data including negative magnetic field is limited by the instrument. Our STM/STS measurements were conducted in Createc LT-STM system with the magnetic coil made by superconducting wire. The magnetic field in the system is at the early stage of the development. The upper limit of downward (negative) magnetic field is only 1.6T, which is insufficient to induce a AFM-FM magnetic phase transition for a few-layer CrI₃ (The critical magnetic field for AFM-FM transition in few-layer CrI₃ is about 1.79 ± 0.10 T according to previous studies as shown in Figure S11). Therefore, we present a complete set of data within upward (positive) magnetic field, wherein the AFM-FM transition can be captured.

2. Can the authors simultaneously measure the magnetic hysteresis loop by magneto-

transport or magneto-optical methods? The combination with the STM data on the same sample will drastically improve the quality and impact of the paper.

Thanks for your suggestion. STM is a local probe for the investigation of atomic-scale structure and properties of this system, while magneto-transport or magneto-optical methods can provide macroscopic-scale properties of this system. These two complementary sets of measurements may not be always necessarily required in the same paper. We feel the macroscopic-scale study of this system using magneto-transport or magneto-optical measurements is beyond the scope of this work. The justifications are provided as follows.

The focus of this manuscript lies in the investigation of atomic structure and magnetism of 2D magnetic insulators *via* tunneling through graphene. By exploiting the transparency of graphene to tunneling electrons, we can probe atomic-scale structure (atomic lattice, interlayer stacking) and electronic properties of few-layer CrI₃ (FL-CrI₃) *via* STM. In addition, AFM-to-FM transition of FL-CrI₃ can be visualized through the magnetic field dependent moiré contrast in the dI/dV maps due to a change of the electronic hybridization between graphene and spin-polarised CrI₃ bands with different interlayer magnetic coupling.

Our results not only provide a general route to probe atomic-scale electronic and magnetic properties of 2D magnetic insulators, but also provide atomic-insights to understand the magnetic properties and AFM-FM transition transitions of few-layer CrI₃ in the previous studies using magneto-transport and magneto-optical methods⁷⁻¹².

3. In figure 3h, the authors analyzed the stacking structure across a single-layer step of CrI₃. Can the author quantify the atomic translation between layers? Also, the error bar is desired.

We thank referee for the valuable suggestion. We quantify the atomic translation between layers using the method reported in the literature [Chen W, Sun Z, Wang Z, et al. *Science*, 2019, **366**(6468): 983-987.]¹³. We first identified the lattices of both upper layer and lower layer, which are represented by red circles and blue circles, respectively (Figure R3b). A statistical analysis shows that the lattice of the lower layer is translated by $\mathbf{L} = (8.35 \pm 0.08)\mathbf{a} + (16.36 \pm 0.06)\mathbf{b}$ with respect to the lattice of the upper layer. Taking the modulus of the translation vector \mathbf{L} , the lower layer is determined to be translated by $(0.35 \pm 0.08)\mathbf{a} + (0.36 \pm 0.06)\mathbf{b}$ with respect to the upper layer, which reveals the monoclinic stacking in exfoliated few-layer CrI₃ at low temperature within the experimental uncertainty.

Actions:

[1] We have included the sentences below in the revised manuscript (page 7, paragraph 1):

“We then identified the lattices of both upper layer and lower layer, which are represented by the red circle and the blue circle, respectively (Figure 3h). A statistical analysis shows the lattice of the lower layer is translated by $\mathbf{L} = (8.35 \pm 0.08)\mathbf{a} + (16.36 \pm 0.06)\mathbf{b}$ with respect to the lattice of the upper layer. Taking the modulus of the translation vector \mathbf{L} , the lower layer is determined to be translated by $(0.35 \pm 0.08)\mathbf{a} + (0.36 \pm 0.06)\mathbf{b}$ with respect to the upper layer, which reveals the monoclinic stacking in exfoliated FL-CrI₃ at low

temperature within the experimental uncertainty. Such a stacking favors the interlayer AFM coupling as predicted by theory^{7,20,24,25}."

[2] We have replaced Figure 3h with Figure R3b and revised description in the main text.

Figure R3. Quantify the atomic translation between layers. (a) The atomic structure of adjacent CrI₃ layers with rhombohedral stacking and monoclinic stacking. The upper (lower) panels are side (top) views. The top view shows the honeycomb lattice formed by Cr atoms (I atoms are removed for clarity), where the center of each hexagon in the upper (lower) layer is indicated by the red (blue) circles. (b) The processed STM image by using edge enhancement filters to better visualize the atomic lattice of both layers. The lattice of the upper (lower) layer are represented by the red (blue) circle. To intuitively show the atomic translation between two layers, a replica of the upper layer lattice (translated by $(8\mathbf{a} + 16\mathbf{b})$ with respect to the original lattice of the upper layer) is shown as the red circle on the lower layer. The red arrow represents the vector $(8\mathbf{a} + 16\mathbf{b})$.

4. In the caption of figure 4a, the red circle shall be noted as well.

Thanks for your suggestion. We have corrected it in our revised manuscript.

Reviewer #3 (Remarks to the Author):

In this paper by Qiu et al., the authors measured the properties of a few layers exfoliated CrI₃ protected with a graphene monolayer. As a result, they were able to avoid the contamination of CrI₃ and measure at the local scale, using STM/STS, this magnetic insulator thanks to graphene's electronic transparency. Although graphene transparency has been previously used to study its interaction with the underlying substrate, the approach showed in this work

opens some interesting possibilities for the STM community to access novel 2D materials. The authors were able to demonstrate the possibility to image the CrI₃ through the graphene layer and measure the gap with dI/dV curves. They also confirmed the monoclinic stack of the layers and the critical magnetic field for the AFM to FM transition. All these results are in good agreement with previous measurements using different techniques and thus I consider this work a nice proof of principle to study different vdW heterostructures with STM in the future.

In general, the manuscript is well written and the arguments are easy to follow. The manuscript also contains sufficient details for others to be able to perform the experiments again.

We thank the reviewer for pointing out that our work offers new possibility for STM to access novel 2D materials, and for providing the constructive suggestions for further improvement. As described below, we have also carefully revised our manuscript and clarified all the points suggested by the reviewer

There are some minor comments that the authors should consider:

-In line 47 they mention that the "application of vdW technology to the STM will dramatically expand the capabilities" of STM. I will not deny the new opportunities that might come from their results, but I have some doubts about the variety of vdW heterostructures that could be built and measured by STM. I guess that using graphene as a capping layer will always be possible, but I don't know if any other material would allow the access to the underlying material properties. This could be a drawback for the final vdW heterostructures suitable for the experiments.

We agree with referee that application of vdW technology to the STM will provide new opportunities for the study of vdW heterostructures.

Such a transparency of graphene in the tunneling process has also been observed for graphene grown on metallic substrate¹⁴. It turns out that the substrate states can extend further beyond the graphene because the graphene's π states are strongly localized by both the large in-plane wave vector of graphene's π states and the small out-of-plane extension of their atomic orbitals¹⁴. We think this is a general picture that can be utilized to probe vdW heterostructures.

In the case of G/CrI₃ heterostructure, our DFT calculations also confirm that the CrI₃ states dominate the simulated STM images at a distance around 4 Å above graphene surface.

Apart from G/2D magnetic insulators, we also performed STM/STS study of a different vdW heterostructure consisting of graphene on a semiconducting monolayer ReSe₂ (G/ML-ReSe₂) (Figure R4a) in our previous work [Qiu, Zhizhan, et al. *Sci. Adv.* **5.7** (2019): eaaw2347.]¹⁵. In this work, we unambiguously resolved the conduction band, valence band and bandgap of the ML-ReSe₂ under graphene (blue curve in Figure 4c). We further reversed the stacking order to fabricate a ML-ReSe₂/G heterostructure (Figure R4b). STS measurements reveal that the electronic band structure of ML-ReSe₂ in these two different heterostructures (G/ML-ReSe₂ and ML-ReSe₂/G) are nearly identical (Figure R4c). Very recently, we also performed STM/STS study of graphene on a new 2D material, which proves the same concept

(unpublished results). All these observations point out that such a concept (the application of vdW technology to the STM) has the general validity.

Therefore, we believe that this method provides an attractive platform to explore different types of vdW heterostructures *via* exploiting the transparency of graphene to tunneling electrons.

Action:

We have included the reference [Qiu, Zhizhan, et al. *Sci. Adv.* **5.7** (2019): eaaw2347.] as ref. 16 in the main text. We also include references [Tesch, J. et al. *Sci. Rep.* **6**, 23439 (2016); Leicht, P. et al. *ACS nano* **8**, 3735-3742 (2014).] as ref. 11-12 in the main text.

Figure R4. Probing the electronic properties of ML-ReSe₂ in graphene based vdW heterostructures. (a) The schematic illustration of a back-gated G/ML-ReSe₂ device. (b) The schematic illustration of a back-gated ML-ReSe₂/G device. (c) The dI/dV spectra on G/ML-ReSe₂ (blue) and ML-ReSe₂/G (red).

-From the dI/dV spectra, the authors associate the Dirac point of graphene with the minimum around 0.13V. If this is true, and the graphene is not strongly interacting with the CrI₃, they could try to measure quasi-particle interference to corroborate it (like in ref. 10 from the main text). For that level of doping and if graphene Fermi velocity is not affected, the k vector at Fermi energy should be somewhere around 0.2 nm⁻¹.

Thanks for your suggestion. Quasi-particle interference (QPI) occurs when there is a scatter center (such as defects) that can scatter the electron from one state to another. In view of referee's suggestion, we attempted to measure QPI over a surface region that contains various defects (Figure R5a). Unfortunately, the QPI observed is so weak, resulting in a non-discernible feature in both dI/dV maps (Figure R5b-d) and corresponding FFT images (Figure

R5f-h). We note the system reported in previous studies of QPI in graphene usually contains various defects with atomically-sharp contrast^{14,16}. In contrast, defects in G/FL-CrI₃ show a slowly varying impurity potential, much larger than the graphene lattice constant (Figure R5b-d). Such a long-range or medium-range perturbation can be neglected as scattering mechanisms [Ando T, Nakanishi T. *J. Phys. Soc. Jpn.*, 1998, **67**(5): 1704-1713.]¹⁷. Therefore, the energy position of Dirac point cannot be inferred by QPI measurement, unlike the case in ref [González-Herrero, H. c. et al. *ACS Nano* **10**, 5131-5144 (2016).]. A further study of the energy position of Dirac point is an interesting topic. However, this requires the use of external gate as a tuning knob or through the transport measurement, which is outside the scope of this paper.

Figure R5. The weak QPI in G/FL-CrI₃. (a) The STM image ($V_s = 0.63 V, I = 1 nA$) showing lots of defects. Defects are outlined by black circles. (b-d) dI/dV maps taken at different sample bias: $V_s = 0.63 V$ (b), $V_s = 0.44 V$ (c) and $V_s = 0.25 V$ (d). The set point of tunneling current is $I_t = 1 nA$. (e) QPI in the first Brillouin zone (BZ) of graphene. The black dot is the reciprocal lattice of graphene. q_F is the Fermi wave vector. The BZ of graphene is indicated by the black dashed hexagon. The ring of diameter $4q_F$ at Γ (red) is related to the intravalley scattering process and the ring of diameter $4q_F$ at K and K' (green) is related to the intervalley scattering process. (f-g) The FFT images of dI/dV maps shown in panel (b-d). The graphene reciprocal lattice is outlined by the white circle.

-How does the charge transfer affect the band structure of CrI₃? The p doping of graphene is mentioned, but the consequences for the CrI₃ are not clear. A way to study it could be changing the graphene/CrI₃ distance in the calculations and observe the evolution of the bands.

Thanks for your suggestion. A direct comparison between the calculated band structure of ML-CrI₃ and G/ML-CrI₃ reveals that the charge transfer will shift the Fermi level into the bottom of CrI₃ conduction band but it has negligible influence on the overall band shape of

CrI₃ [Zhang, J. et al. *Phys. Rev. B* **97**, 085401 (2018)]¹⁸ (this reference has been cited as ref. 22 in the main text).

As discussed in the main text (page 5, paragraph 1), there is some discrepancy between predicted band structures with the respect to the Fermi level and experimental data, which can be attributed to the following two factors: (i) A suspending G/CrI₃ model (without substrate) is calculated in the DFT, while G/CrI₃ sample is placed on a graphite substrate. There may be charge transfer between G/CrI₃ and graphite substrate as mentioned in the main text. (ii) it is likely that the presence of defects in CrI₃ also introduce the doping and change the overall Fermi level.

The impact of the interface distance on G/CrI₃ bands has been explored in the previous work [Zhang, J. et al. *Phys. Rev. B* **97**, 085401 (2018)]¹⁸. Our results show good agreement with the calculated band structure of G/CrI₃ at equilibrium interface distance in this Ref [Zhang, J. et al. *Phys. Rev. B* **97**, 085401 (2018)]. In addition, the authors also found that as a reduction of interface distance, the FM CrI₃ substrate will induce a very large magnetic exchange field and Rashba SOC in graphene, which may potentially lead to Chern insulating state. In addition, a decrease of the interface distance will move the Fermi level from the position near the bottom of the conduction bands to the top of the valence bands, indicating the variation from Ohmic contacts to Schottky contacts at the G/CrI₃ interface. The theoretical results reported by Zhang, J. et al. point out a potential route to realize Chern insulating state in graphene by compressing graphene/CrI₃. Such a calculation is not closely related to our experimental results.

-Did the authors observe any change of visualization from graphene to CrI₃ depending on the current? One would expect the possibility to visualize the graphene layer or the CrI₃ at some voltages just changing the value of the tunneling current.

Thanks for your comment. It is noted that the visualization of graphene or CrI₃ lattice mainly depends on the sample bias rather than tunnelling current. Figure R6a-d shows the STM images taken at $V_s = 1V$ by varying the tunneling current from 0.1 nA to 1 nA. All these STM images resolves CrI₃ lattice, independent of tunnelling current. That is because the electronic states probed are deep into the CrI₃ conduction band and thus the CrI₃ states have a major contribution to the tunneling current.

A smaller positive bias ($V_s = 0.44V$) probes the electronic states near the CrI₃ conduction band edge as well as graphene states. In this case, the weight of integrated graphene states and integrated CrI₃ states may be comparable so that both graphene lattice and CrI₃ lattice can be resolved (Figure R5e-h). A variation of tunneling current from 0.1 nA to 1 nA has a limited influence on the visualization of graphene lattice. An increase of tunneling current makes the graphene (CrI₃) lattice slightly more (less) prominent.

Action:

[1] We have included Figure R6 with the description in the revised supplementary information to present STM images recorded at varied tunnelling currents (sample bias is kept the same for each set of data) (Figure S2 in S1).

Figure R6. STM images of G/FL-CrI₃/Gr recorded at varied tunneling current. (a-d) STM images ($V_s = 1V$) recorded at varied tunneling current from 0.1 nA to 1 nA. Note the STM images shown could be taken on different areas. (e-h) STM images ($V_s = 0.44V$) recorded at varied tunneling current from 0.1 nA to 1 nA.

-Regarding the magnetic field dependent moiré measurements, did the authors measure any other size of moiré? I would expect that for small moirés where the difference in interaction between the lower and higher region is going to be smaller, the contrast difference would be smaller and therefore addressing the value for the AFM-FM transition will be complicated. How much control to obtain different moirés do the authors have?

We agree with referee that moiré modulation is likely to be weaker for a smaller moiré superlattice, as suggested by previous studies in other moiré systems^{19,20}. We have performed STM/STS study of a moiré superlattice with a periodicity of 1.84 nm (Figure R7a). Both spatially dependent dI/dV spectra (Figure R7c) and dI/dV map (Figure R7b) show a non-discernible spatial moiré modulation on the LDOS at zero magnetic field. Therefore, a negligible moiré modulation in a small moiré superlattice makes it impossible to address AFM-FM magnetic phase transition via the visualization of magnetic field dependent moiré contrast in dI/dV maps.

Figure R7. The non-discernible moiré modulation in a small moiré superlattice. (a) a small moiré superlattice with a periodicity of 1.84 nm as shown in the STM image ($V_s =$

0.4V, $I_t = 0.5nA$). The unit cell of moiré superlattice is outlined by the black curve. (b) The corresponding dI/dV map ($V_s = 0.4V, I_t = 0.5nA$). (c) Spatially dependent dI/dV spectra taken at different locations. The tip position where we took the blue (red) dI/dV spectrum is indicated by the blue (red) cross in panel (a-b).

-Although ref. 10 from the main text was the first one trying to understand the mechanism behind the transparency of graphene, the works on graphene on Au(111) and Ag(111) from the group of Prof. Mikhail Fonin should probably be included

(<https://doi.org/10.1038/srep23439> , <https://doi.org/10.1021/mn500396c>).

Thanks for your suggestion. We have included the references [Tesch, J. *et al. Sci. Rep.* **6**, 23439 (2016); Leicht, P. *et al. ACS Nano.* **8**, 3735-3742 (2014).]^{21,22} as ref.11-12 in the revised manuscript.

-In line 29-30 the authors mention the "lack of magnetic sensitivity of traditional STM". I would say that spin polarized STM is a well-established technique nowadays and maybe I would not include it in the list with "absence of a conducting path and their extreme air sensitivity".

We agree with referee. We have revised it accordingly.

Action: We have revised the sentence in the manuscript (page 2, paragraph 1):

“Further progress hinges on deep understanding of electronic and magnetic properties of 2D magnets at the atomic scale. Although local electronic properties can be probed by scanning tunneling microscopy/spectroscopy (STM/STS), its application to investigate 2D magnetic insulators remains elusive due to absence of a conducting path and their extreme air sensitivity¹⁻³.”

- The authors should include the tunneling current values in the figure captions.

Thanks for your suggestion. We have included the tunneling current values in the figure captions in the revised manuscript.

Reference:

- 1 Jiang, S., Shan, J. & Mak, K. F. Electric-field switching of two-dimensional van der Waals magnets. *Nat. Mater.* **17**, 406-410 (2018).
- 2 Jiang, S., Li, L., Wang, Z., Mak, K. F. & Shan, J. Controlling magnetism in 2D CrI₃ by electrostatic doping. *Nat. Nanotech.* **13**, 549-553 (2018).
- 3 Jiang, S., Li, L., Wang, Z., Shan, J. & Mak, K. F. Spin tunnel field-effect transistors based on two-dimensional van der Waals heterostructures. *Nat. Electron.* **2**, 159-163 (2019).
- 4 Huang, B. et al. Layer-dependent ferromagnetism in a van der Waals crystal down to the monolayer limit. *Nature* **546**, 270-273 (2017).

- 5 Thiel, L. et al. Probing magnetism in 2D materials at the nanoscale with single-spin
microscopy. *Science* **364**, 973-976 (2019).
- 6 McCreary, A. et al. Distinct magneto-Raman signatures of spin-flip phase transitions
in CrI₃. *Nat. Commun.* **11**, 1-8 (2020).
- 7 Li, T. et al. Pressure-controlled interlayer magnetism in atomically thin CrI₃. *Nat.*
Mater. **18**, 1303-1308 (2019).
- 8 Song, T. et al. Switching 2D magnetic states via pressure tuning of layer stacking. *Nat.*
Mater. **18**, 1-5 (2019).
- 9 Song, T. et al. Giant tunneling magnetoresistance in spin-filter van der Waals
heterostructures. *Science* **360**, 1214-1218 (2018).
- 10 Klein, D. R. et al. Probing magnetism in 2D van der Waals crystalline insulators via
electron tunneling. *Science* **360**, 1218-1222 (2018).
- 11 Zhong, D. et al. Van der Waals engineering of ferromagnetic semiconductor
heterostructures for spin and valleytronics. *Sci. Adv.* **3**, e1603113 (2017).
- 12 Wang, Z. et al. Very large tunneling magnetoresistance in layered magnetic
semiconductor CrI₃. *Nat. Commun.* **9**, 1-8 (2018).
- 13 Chen, W. et al. Direct observation of van der Waals stacking-dependent interlayer
magnetism. *Science* **366**, 983-987 (2019).
- 14 González-Herrero, H. c. et al. Graphene tunable transparency to tunneling electrons: a
direct tool to measure the local coupling. *ACS Nano* **10**, 5131-5144 (2016).
- 15 Qiu, Z. et al. Giant gate-tunable bandgap renormalization and excitonic effects in a
2D semiconductor. *Sci. Adv.* **5**, eaaw2347 (2019).
- 16 Rutter, G. M. et al. Scattering and interference in epitaxial graphene. *Science* **317**,
219-222 (2007).
- 17 Ando, T. & Nakanishi, T. Impurity scattering in carbon nanotubes—absence of back
scattering. *J. Phy. Soc. Jpn.* **67**, 1704-1713 (1998).
- 18 Zhang, J. et al. Strong magnetization and Chern insulators in compressed
graphene/CrI₃ van der Waals heterostructures. *Phys. Rev. B* **97**, 085401 (2018).
- 19 Decker, R. et al. Local electronic properties of graphene on a BN substrate via
scanning tunneling microscopy. *Nano Lett.* **11**, 2291-2295 (2011).
- 20 Alexeev, E. M. et al. Resonantly hybridized excitons in moiré superlattices in van der
Waals heterostructures. *Nature* **567**, 81-86 (2019).
- 21 Tesch, J. et al. Structural and electronic properties of graphene nanoflakes on Au (111)
and Ag (111). *Sci. Rep.* **6**, 23439 (2016).
- 22 Leicht, P. et al. In situ fabrication of quasi-free-standing epitaxial graphene
nanoflakes on gold. *ACS Nano* **8**, 3735-3742 (2014).